# RhoGEF Ect2 supports RhoA activity at cell–cell junctions through desmoplakin

Hoda Zarkoob[1],*, Chen Y Kam[1],*, Jennifer L Koetsier[1], Erin McCarthy[1] , Avinash Jaiganesh[1], David P Kelsell[2] , Farah Sheikh[3], Lisa M Godsel[1] , Kathleen J Green[1]

**Desmoplakin (DP) is an essential component of the desmosomal adhesion complex, tethering intermediate filaments to sites of intercellular adhesion to confer mechanical integrity to tissues. As a frequent target for mutation in cardiocutaneous syndromes that vary widely in phenotype, DP's roles as a signaling hub are rapidly emerging. Here, we identify the RhoGEF Ect2 as a previously unappreciated component of intercellular junctions in close association with DP. DP promotes the localization of Ect2 to keratinocyte desmosomes and cardiac intercalated discs, where it maintains active RhoA (Rho-GTP) at the membrane. We demonstrate that Ect2 activity is regulated by PKC in a DP-dependent manner in cardiac myocytes. Finally, a truncated form of DP expressed in patients with Carvajal syndrome associated with severe cardiocutaneous defects is impaired in its ability to bind and localize Ect2 to cell junctions in cardiomyocytes and patient keratinocytes. Our findings delineate an important relationship between a component of the desmosome and a critical regulator of actin cytoskeletal remodeling that could have widespread implications for understanding cardiac and cutaneous health and disease pathogenesis.**

## Introduction

Cell–cell junctions are vital sensors and integrators of mechanical forces experienced on the molecular, cellular, and tissue scale. In addition to extrinsic forces experienced by the tissue, cell–cell junctions are also responsible for mediating intrinsic forces generated by cells of the tissue themselves, which are generally tensile forces that are generated when adhesion is coupled to the contractile actomyosin network (1, 2, 3). One of the best characterized modulators of actomyosin contractility is the RhoA GTPase signaling pathway (4, 5). Activation of RhoA leads to the direct modulation of its various downstream effectors, which have the cumulative outcome of promoting actin polymerization and myosin contractility. The importance of RhoA as a crucial regulator of junctional integrity and cellular contractility is now established (6, 7, 8). Properly tuned RhoA activity is critical to maintain a balance of contractile activity at the cell cortex necessary for establishment and maintenance of polarized epithelial cell functions, and interfering with Rho signaling is associated with disease pathogenesis (9).

Given the importance of RhoA in epithelia, it is surprising how little is known about junctional RhoA or the upstream factors that control its spatiotemporal activation in cardiac cell–cell junctions. Furthermore, the extent to which these pathways are shared with those in epithelia is unknown. Unlike the multinucleated syncytium of skeletal muscle, cardiomyocytes are organized as discrete rod-shaped units that interconnect along their longitudinal axis via highly specialized intercellular junctions, referred to as intercalated discs (IDs) (10). IDs represent the primary site of intercellular adhesion and electrochemical coupling between cardiomyocytes and can carry out these functions because of their unique junctional composition. Within IDs, components of adherens junctions (AJs), desmosomes, and gap junctions are found in close proximity to each other and are able to form hybrid junctions that have been termed "area composita," in which components of desmosomes and AJs are intermixed (11). This is particularly important as IDs represent the primary site of force transmission between cardiomyocytes whereby the contractile forces generated by sarcomeres are transmitted to neighboring cardiomyocytes via these specialized junctions (10).

RhoA signaling has emerged as a somewhat controversial player in cardiac pathology (12); however, despite the importance attributed to this contractile signaling modulator in epithelia, its role in the ID is understudied. This is particularly notable given that intercellular junctions are major targets in disorders involving both skin and heart, including arrhythmogenic cardiomyopathy and related syndromes like Carvajal disease. 50% of patients presenting with these disorders harbor pathogenic variants of molecules in cadherin-based intercellular junctions called desmosomes, which physically and functionally network with AJs.

[1]Department of Pathology, Northwestern Feinberg School of Medicine, Chicago, IL, USA   [2]Blizard Institute, Faculty of Medicine and Dentistry, Queen Mary University of London, London, England   [3]School of Medicine, Department of Medicine, University of California-San Diego, La Jolla, CA, USA

Correspondence: l-godsel@northwestern.edu; kgreen@northwestern.edu
*Hoda Zarkoob and Chen Y Kam contributed equally to this work

Although heart and skin share the property of being mechanically stressed tissues, the underlying mechanisms linking specific variants to the heterogeneous phenotypes in these disorders are unknown.

Our laboratory previously reported that desmosome components play an important role along with their attached intermediate filaments (IFs) in balancing the contractile activity mediated by AJs and their associated cortical actin (13, 14). One such component is the cytolinker protein desmoplakin (DP), which functions to tether IFs to sites of cell–cell adhesion, providing mechanical integrity to the tissues in which it resides. The expression of a mutant version of DP lacking its IF binding domains (DPNTP) resulted in a depletion of actin-dependent cellular forces that were associated with a concurrent reduction of active RhoA at epithelial cell junctions (13). This finding, along with previous evidence that plakophilin 2, a desmosomal component that is vital for DP's localization to cell junctions, is involved in regulating RhoA GTPase junctional signaling (14), led us to hypothesize that DP could be acting as a scaffold for regulators of junctional RhoA GTPase signaling.

The class of proteins responsible for RhoA activation is the guanine nucleotide exchange factors (GEFs) that catalyze the exchange of GDP for GTP on Rho proteins, resulting in the activation of its GTPase function (15). As such, the subcellular localization of specific GEFs is an important determinant of Rho GTPase activation at corresponding sites within a cell. The RhoGEF Ect2 has previously been documented to be localized to cell–cell junctions of interphase cells at steady state in models of simple epithelia (16). In simple epithelia, Ect2's junctional localization was found to be dependent upon the AJ protein α-catenin and, once present at cell borders, forms a complex with α-catenin and its binding partner E-cadherin (16). The junctional presence of Ect2 was further demonstrated to be crucial for the maintenance of an active pool of RhoA at sites of cell–cell adhesion and the promotion of junctional tension. However, it has yet to be established whether Ect2 can be found at cellular junctions in more complex tissues such as the stratified epithelium of the epidermis or the myocardium and to what extent it could be playing a role in modulating the junctional tension and cellular forces within these tissues.

In this study, we identify the RhoGEF Ect2 as a previously unappreciated component of the cardiac ID in close association with the desmosomal protein DP. We further demonstrate that DP promotes the localization of Ect2 to cardiac junctions in both in vitro and in vivo models, independent of the AJ machinery that it uses in simple epithelial models. Depletion of either DP or Ect2 was found to be sufficient to significantly perturb the maintenance of an active pool of RhoA that is present at steady state at cardiac cell junctions. Finally, we demonstrate that a truncated form of DP harboring a patient mutation that results in severe cardiocutaneous defects is impaired in its ability to associate with and localize Ect2 to cell junctions in cardiomyocytes and keratinocytes isolated from patients. Our findings delineate an important regulatory relationship between a component of the desmosome and a critical regulator of cytoskeletal remodeling that could have widespread implications for our understanding of cardiac and cutaneous health and disease progression.

## Results

### Ect2 is a novel component of the cardiac intercalated disc depending on desmoplakin

We previously showed that the expression of DP mutants that uncouple the IF network from the desmosomal complex impairs the junctional distribution and activation of the small GTPase RhoA (13). As it is well established that spatiotemporal control of RhoA activation is heavily dependent upon the localization of GEFs to specific sites within the cell, we hypothesized that DP acts as a scaffold for a RhoGEF at cardiac cell junctions. Previous studies identified the RhoGEF Ect2 as a critical regulator of junctional RhoA GTPase activation and junctional tension in simple epithelial cells, and as such, Ect2 is a plausible candidate regulator of RhoA activity in cardiac junctions.

We first tested for the presence of Ect2 at cardiac cell junctions using the well-established primary cell culture model of neonatal rat ventricular cardiomyocytes (NRVCMs). Immunostaining of freshly isolated NRVCMs for Ect2 and ID components DP and α-catenin, followed by structured illumination microscopy (SIM) imaging, showed that Ect2 is indeed found to be enriched at cardiac cell junctions (Figs 1A and S1A). Having shown the presence of Ect2 at cardiac junctions, we next addressed whether this localization is conserved in vivo. To this end, we carried out immunostaining of Ect2 on murine and human cardiac tissue and observed a robust Ect2-positive signal colocalizing with DP at murine and human IDs (Figs 1B and S1B and C). To corroborate association of Ect2 and DP at the plasma membrane, we carried out proximity ligation assay (PLA) analysis of DP and Ect2 in NRVCMs. This method allows for the detection of proteins that are in close proximity to one another (40–100 nm) and is frequently used to detect in situ protein–protein interactions. Complementing the PLA analysis, immunoprecipitation of DP followed by blot back with an Ect2 antibody in the murine cardiac cell line, HL-1s, and in human keratinocytes showed that Ect2 was specifically pulled down with DP, compared with the IgG control (Fig S1D and E). To assess the necessity of DP for localization of Ect2 at IDs, NRVCMs were treated with adenovirus harboring either control (nontargeting) or DP KD shRNA constructs followed by PLA analysis. Control NRVCMs showed a robust DP-Ect2 PLA signal that significantly decreased in DP KD cells, indicating that this signal was specific to the DP-Ect2 antibody pairing (Fig 1C and C').

These data indicate that Ect2 is a novel component of the ID complex in close proximity to DP, and associates with DP in cardiac and epithelial cells. Given the importance of spatial distribution of GEFs in localizing and activating RhoA, we carried out further experiments to elucidate the role played by the DP-Ect2 complex in cardiac function and homeostasis.

### DP is required for the localization of Ect2 to cardiac cell junctions

Toward understanding the importance of DP for Ect2 localization, we first asked whether DP is required for Ect2's recruitment to cardiac cell junctions. NRVCMs were treated with adenovirus harboring control, DP KD, or Ect2 KD shRNA constructs followed by

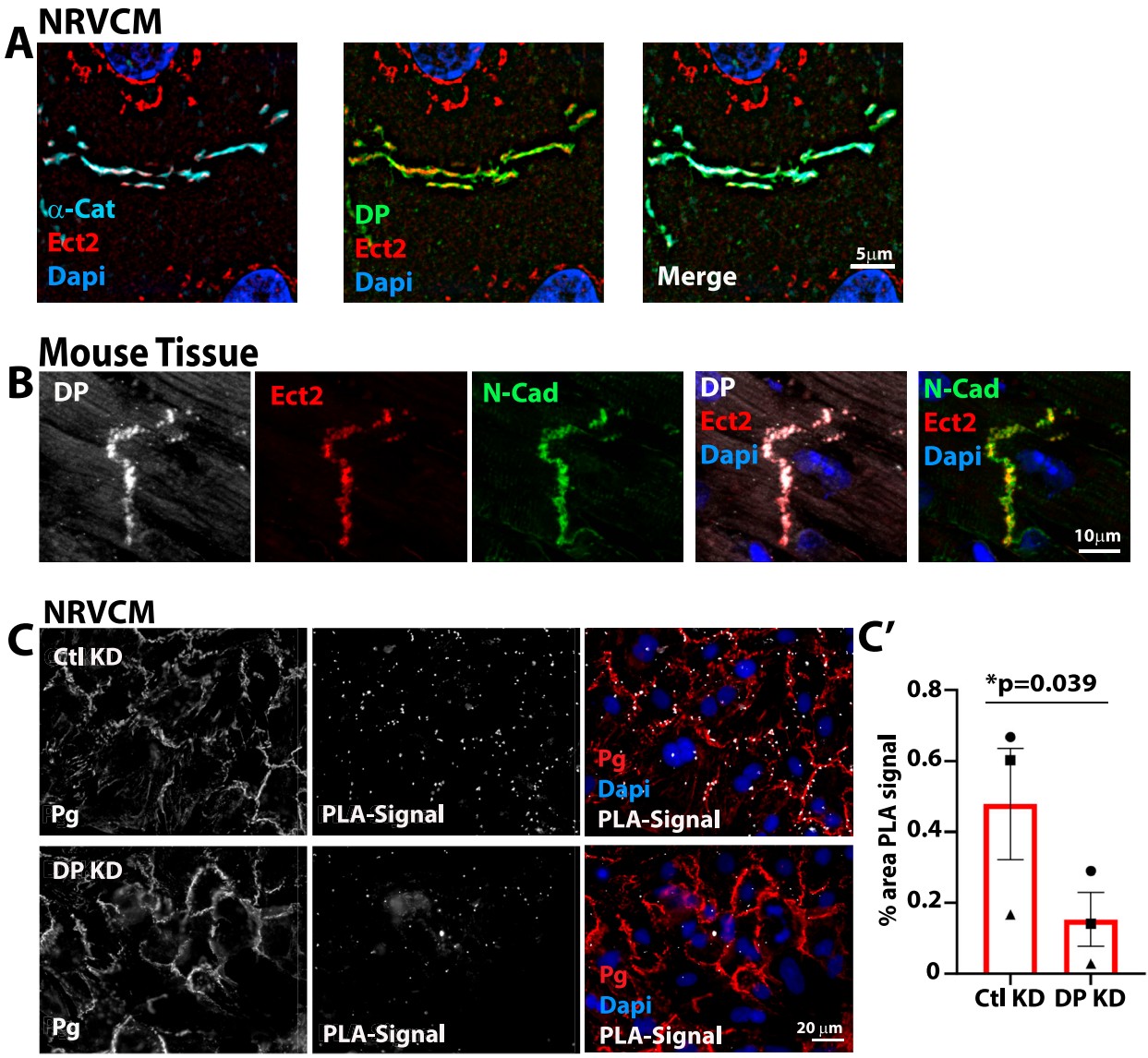

**Figure 1.  Desmoplakin and the RhoGEF Ect2 colocalize at the intercalated disc.**
**(A)** Isolated cardiac myocytes (NRVCMs) were prepared for immunofluorescence and labeled for desmoplakin (DP), Ect2, and α-catenin and imaged using structured illumination microscopy. Orthogonal views can be found in Fig S1A. Scale bar = 5 μm. **(B)** Sections from control mouse hearts were fixed and stained for DP, Ect2, and N-cadherin (N-Cad) and imaged using the AxioVision Z1 system (Carl Zeiss) with Apotome slide module. Scale bar = 10 μm. **(C)** Proximity ligation assay (PLA) was performed on NRVCMs in control (Ctl KD) and DP knockdown (DP KD) conditions using antibodies directed against DP and Ect2. Scale bar = 20 μm. **(C')** Quantification of each of three paired experiments was performed, showing significantly more PLA spots in Ctl KD conditions, indicating close proximity of DP and Ect2 (n = 3; PLA signal area for 17 control and 14 DP KD fields, paired $t$ test,*$P$ = 0.039). DAPI is used to stain nuclei in (A, B, C).

immunofluorescence staining of DP and Ect2. The desmosomal component plakoglobin (Pg) was also labeled as a marker of cell junctions as we have previously shown that Pg junctional localization is not significantly affected by depleting DP in cardiac cells (17). Efficacy of the shRNA directed toward DP and Ect2 was confirmed by the loss of immunofluorescence signal at cell junctions (Fig 2A).

Depletion of DP in NRVCMs resulted in a significant decrease in Ect2 signal at cardiac cell junctions identified by Pg staining (Fig 2A). There is also an Ect2 fluorescent signal marking the periphery of the nucleus, which was only partially affected by Ect2 KD

shRNA. Whether this remaining nonjunctional pool represents specific signal, not targeted by the shRNA construct, is unclear. To verify that loss of junctional Ect2 is specifically due to DP KD, we re-expressed exogenous DP by introducing an adenovirus harboring a DPII-GFP construct in the background of DP KD. DPII is a naturally occurring isoform of DP that lacks a portion of its central rod domain (6.8 versus 8.6 kb), which allows us to overcome the technical limits in size for adenoviral packaging and transduction. DPII expression in a DP-deficient background was sufficient to significantly restore Ect2's localization to cardiac cell junctions (Fig 2A and A').

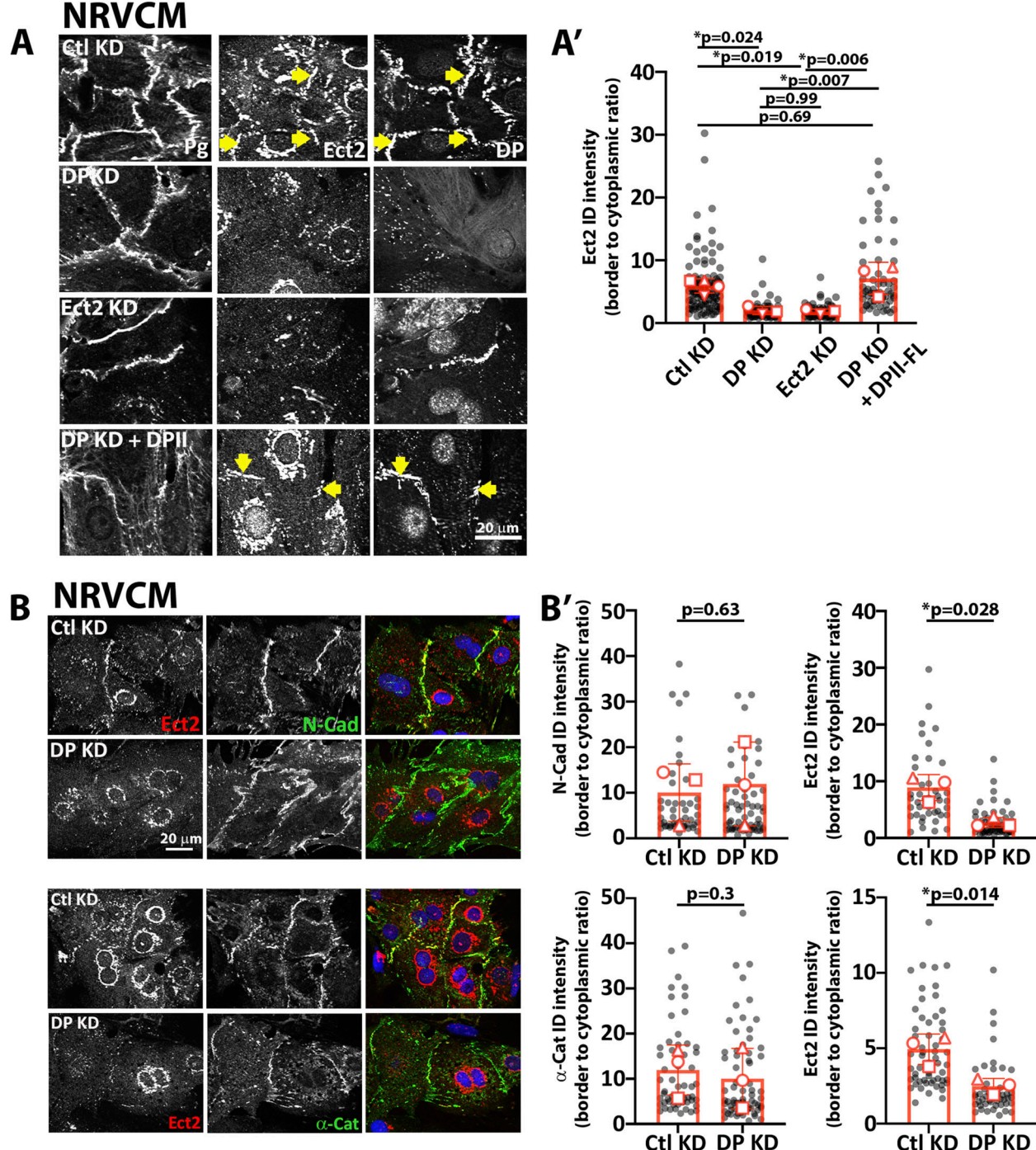

**Figure 2.  DP is required for the localization of Ect2 to cardiac cell junctions in vitro.**
**(A)** Isolated cardiac myocytes (NRVCMs) were treated with adenovirus encoding scrambled control (Ctl), DP, or Ect2 knockdown (KD) oligonucleotides, and then stained for DP and Ect2, as well as plakoglobin (Pg), to mark cell–cell borders. In addition, DP KD was rescued by expression of full-length desmoplakin II (DPII-FL). Yellow arrows indicate cell junctions containing DP and Ect2. **(A′)** Quantification of fluorescence intensity at the cell–cell junctions showed that Ect2 junctional staining was lost in the absence of DP and could be rescued by ectopic DPII expression (n = 3; ROIs: Ctl = 110, DP KD = 71, Ect2 KD = 67, and DP KD + DPII rescue = 58, repeated-measures one-way ANOVA with Tukey's multiple comparisons test, *$P < 0.05$). **(B)** NRVCMs were treated with adenovirus encoding scrambled control (Ctl) or DP KD oligonucleotides and stained for Ect2 and the adherens junction proteins N-cadherin (N-Cad) or $\alpha$-catenin ($\alpha$-Cat). **(B′)** Quantification of fluorescence intensity at the cell–cell junctions showed that neither N-cadherin nor $\alpha$-catenin was significantly changed with DP KD, but Ect2 border intensity significantly decreased (n = 3; junction measurements: N-Cad/Ect2 = 44 Ctl, 62 DP KD; and $\alpha$-catenin/Ect2 = 56 Ctl 68 DP KD, paired $t$ test, $P \leq 0.05$).

A previous study in simple epithelial cells reported that Ect2 forms a complex with the AJ components E-cadherin and α-catenin and colocalizes with AJs (16). Furthermore, the disruption of AJs via KD of α-catenin was sufficient to disrupt Ect2's localization to epithelial borders. To determine whether the failure of Ect2 to localize to cell borders under DP KD conditions was due to concurrent perturbations to AJs, we sought to assess the status of the cardiac AJ components, α-catenin and N-cadherin, via immunofluorescence analysis in a DP-deficient background. Ect2, N-cadherin, and α-catenin were present at cell borders in control NRVCMs as expected; however, even though DP KD led to a significant reduction in junctional Ect2, there were no significant changes in N-cadherin or α-catenin at corresponding junctions (Fig 2B and B′).

We then asked whether DP-dependent junctional localization of Ect2 is conserved in vivo, using a previously reported murine cardiac conditional DP knockout (*Dsp* cKO) model (18). This model was generated by crossing *Dsp*-floxed mice (*Dsp*^flox/flox^) with the well-established cardiomyocyte-specific ventricular myosin light chain-2-Cre recombinase (MLC2v-Cre) knock-in mice. Cardiac sections from 8-wk-old WT and surviving *Dsp* cKO mice were immunostained with antibodies directed against DP and Ect2, and Pg was used as a marker of IDs. As previously reported, DP staining was abolished in many of the IDs in *Dsp* cKO hearts. Retention of DP in some of IDs was not unexpected because of the level of cardiomyocyte targeting efficiency for the MLC2v-Cre line (Fig 3A). Overall, *Dsp* cKO cardiac sections displayed a significant reduction in Ect2 signal at IDs (Figs 3A and A′ and S2A). In contrast to our in vitro observations, the Ect2 signal was occasionally observed at DP-deficient IDs, raising the possibility that other junctional proteins may aid in Ect2 localization in vivo where junctions have been long established in the mouse ventricle (Fig 3A, boxes). As discussed below, however, active RhoA depended on the presence of DP at IDs in vivo. Consistent with our in vitro studies, neither N-cadherin nor α-catenin ID intensity was significantly affected in *Dsp* cKO hearts (Fig 3B, B′, C, and C′).

### DP loss results in reduction of activated RhoA at cardiac cell junctions

A role of RhoA signaling in the regulation of cardiac development and homeostasis is well documented, if somewhat controversial (12, 19, 20). However, the role and regulation of junctional-specific RhoA in cardiac systems have gone largely unexplored. To assess the status of RhoA and its activation in cardiac junctions, we employed the well-established TCA fixation method that allows for the visualization of junctional-specific endogenous RhoA by immunofluorescence (21, 22). Using this fixation method, we assessed the distribution of active RhoA in NRVCMs with an antibody that specifically recognizes the GTP-bound active form of RhoA (Rho-GTP). In addition to the expected cytoplasmic distribution of RhoA, we detected a concentration of the RhoA-GTP signal along the cell–cell junctions of control NRVCMs as visualized by costaining with Pg (Fig 4A). Furthermore, DP or Ect2 KD resulted in a significant reduction of junctionally localized RhoA-GTP, confirming the role of these proteins in the maintenance of the junctional-specific pool of active RhoA in cardiac cells (Fig 4A and A′). We also noted a

trend toward reduction of DP staining at cell–cell junctions in Ect2 KD NRVCMs, whereas Pg levels at junctions did not vary (Figs 4A and S2B and B′). Although the underlying basis of this reduction is not clear, it is plausible that loss of RhoA-mediated contractility could decrease junctional DP stability. A full-length DPII-GFP construct expressed in a DP-deficient background at levels that matched WT DP at junctions was sufficient to restore the RhoA-GTP signal back to that of control levels (Fig 4A and A′).

We also tested whether these observations held true in vivo, using the murine model of cardiac *Dsp* cKO. Immunofluorescence analysis confirmed that the RhoA-GTP signal is concentrated at mature IDs of WT mouse heart sections (Fig 4B and insets). Supporting our in vitro findings, *Dsp* cKO cardiac sections displayed a significant reduction of the ID-specific RhoA-GTP signal at Pg-labeled IDs where the DP signal was absent or reduced ("DP low" IDs) (Fig 4B and insets, Figs 4B′ and S2C), without a significant change in Pg levels (Fig S2D). These data show that DP is required for the maintenance of an active pool of RhoA-GTP at cardiac cell junctions. When considering the observed retention of Ect2 at some DP-deficient IDs, these data suggest that the presence of DP is required for Ect2 activity. To test whether DP controls Ect2's catalytic GEF function toward RhoA, we employed an assay designed to precipitate the active pool of GEFs from HL-1 cardiac cell lysates. This technique takes advantage of a single amino acid mutation in RhoA (G17A) that is stabilized in a nucleotide-free state, which has a high affinity toward active RhoA-specific GEFs (23). We observed that DP KD resulted in a significant reduction of Ect2 precipitated by RhoA (17A) beads compared with control lysates (Fig 4C and C′). This result confirms that DP is an important regulator of Ect2's ability to catalyze the exchange of GDP for GTP on RhoA.

### DP regulates Ect2 activity via PKC-mediated phosphorylation

We next set out to determine the mechanism by which DP regulates activation of Ect2. It was previously reported that protein kinase C (PKC)–dependent phosphorylation of Ect2 up-regulates its activity (24, 25). Furthermore, our laboratory previously showed that DP forms a complex with PKCα and its desmosomal binding partner, plakophilin 2 (26). As such, we tested the hypothesis that DP could be acting as a scaffold for PKC at cardiac cell junctions to allow for phosphorylation and activation of Ect2.

We first assessed whether Ect2's activation state in cardiac cells is responsive to alterations in PKC kinase activity levels. To do so, we leveraged the Rho (G17A) pull-down assay to determine Ect2's activation levels in the presence of pharmacological modulation of PKC activity. Treatment of HL-1 cells with the broad-spectrum PKC inhibitor, BIM1, led to a significant decrease in Ect2 activity compared with DMSO vehicle treatment (Fig 5A and 5A′). Conversely, after treatment with the well-established PKC activator, PMA, Ect2 activation status was significantly up-regulated in HL-1 cells compared with DMSO-treated controls, suggesting that Ect2 activation can be modulated by PKC kinase activity.

To determine whether DP plays a role in regulating PKC-mediated phosphorylation of Ect2, we next carried out immunoprecipitation (IP) of endogenous Ect2 in control and DP KD NRVCMs

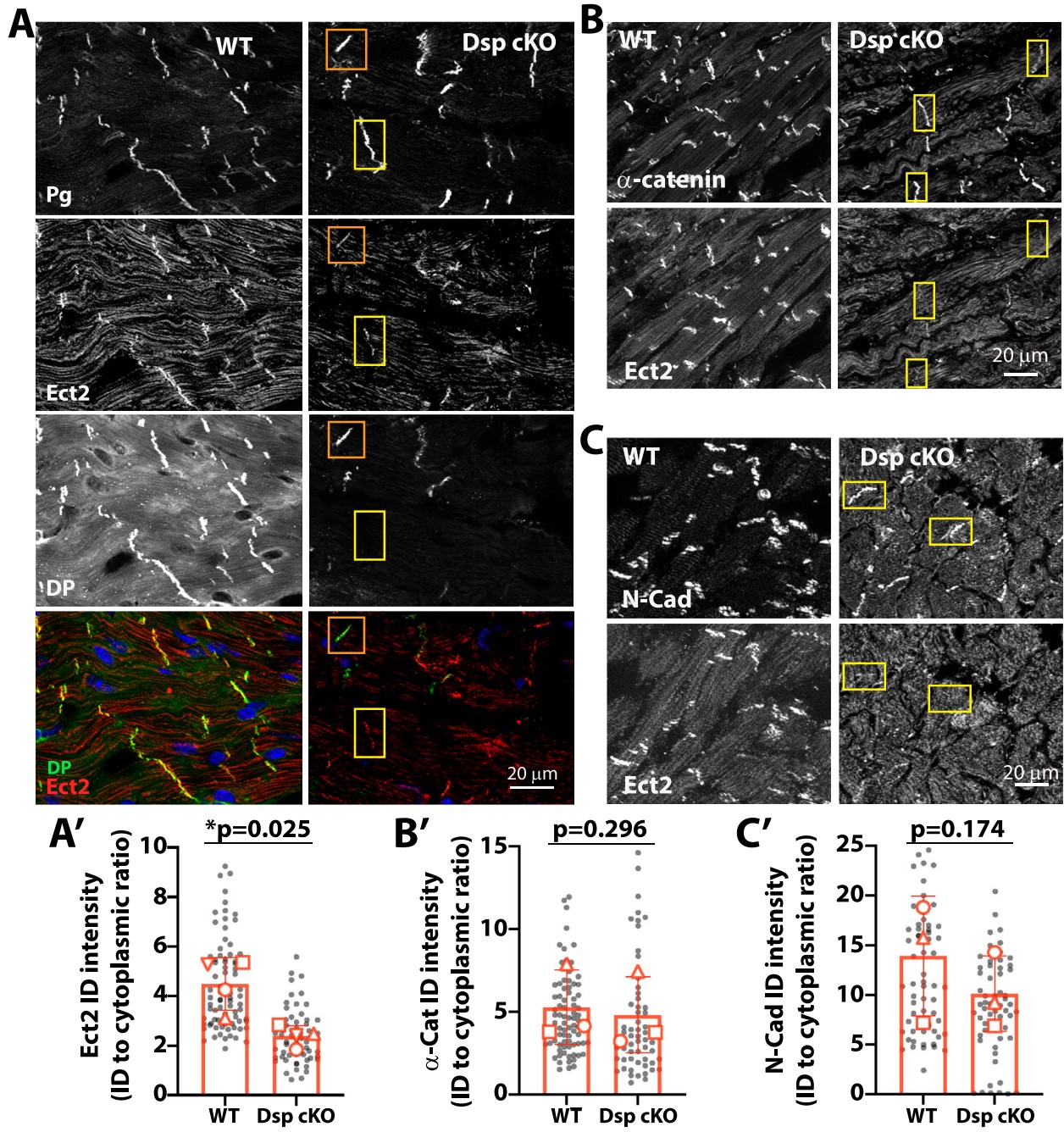

**Figure 3. DP supports Ect2 localization at cardiac cell junctions in vivo.**

**(A)** Sections from control and *Dsp* cKO mouse hearts were fixed and triple-labeled for DP, Ect2, and Pg to mark IDs. **(A′)** Quantification of fluorescence intensity at IDs showed a significant decrease in Ect2 at *Dsp* cKO IDs. Orange and yellow boxes indicate examples of junctions where DP is present or absent, respectively (n = 4; ROIs: 79 WT IDs, 61 *Dsp* cKO IDs, paired *t* test, *P* = 0.025). **(B)** Sections from WT and *Dsp* cKO mouse hearts were fixed and double-labeled for α-catenin and Ect2. Yellow boxes indicate junctions containing α-catenin where Ect2 is absent. **(B′)** Quantification of fluorescence intensity at IDs showed no significant decrease in α-catenin at *Dsp* cKO IDs (n = 3; ROIs: 79 WT IDs, 58 *Dsp* cKO IDs, paired *t* test). **(C)** Sections from control and *Dsp* cKO mouse hearts were fixed and double-labeled for N-cadherin and Ect2. Yellow boxes indicate junctions containing N-cadherin where Ect2 is absent. **(C′)** Quantification of fluorescence intensity at IDs showed no significant decrease in N-cadherin at *Dsp* cKO IDs (n = 3; ROIs: 65 WT IDs, 60 *Dsp* cKO IDs, paired *t* test). Scale bars = 20 μm.

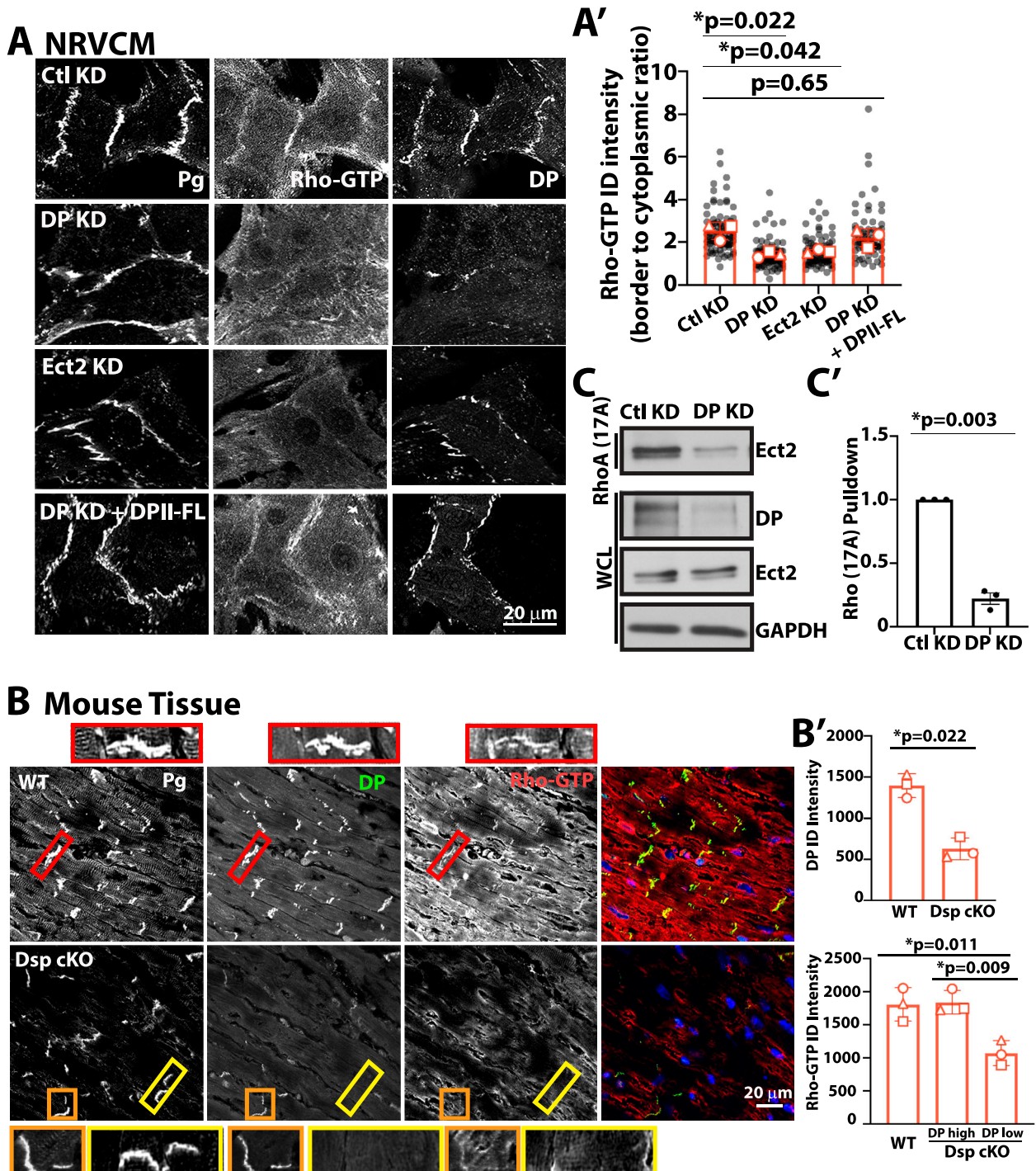

**Figure 4. Active RhoA at cardiac cell junctions requires the expression of DP and colocalizes with DP at IDs.**
**(A)** Isolated cardiac myocytes (NRVCMs) were treated with scrambled control (Ctl), DP, or Ect2 knockdown (KD) oligonucleotides, and then stained for plakoglobin (Pg) to mark cell–cell borders, as well as an antibody that recognizes active RhoA (Rho-GTP). Active RhoA cell junction localization is lost upon either DP or Ect2 silencing, and its border localization is restored by ectopic expression of DPII-FL. **(A′)** Quantification of Rho-GTP intensity in (A) (n = 3; ROIs: Ctl KD = 92 IDs, DP KD = 85 IDs, Ect2 KD = 80 IDs, DP KD + DPII-FL rescue = 70 IDs, repeated-measures one-way ANOVA with Tukey's multiple comparisons test). **(B)** Sections from control and *Dsp* cKO mouse hearts were fixed and triple-labeled for Pg to mark IDs, DP, and active RhoA. **(B′)** Quantification of DP border intensity in IDs of the WT versus *Dsp* cKO animals and Rho-GTP intensity at IDs of the WT and *Dsp* cKO animals showed a significant decrease in active RhoA at IDs with low DP localization compared with WT (DP low, *P = 0.011) or low DP localization IDs compared with IDs that retain DP (DP high IDs, *P = 0.009) (n = 3; 5–10 fields per condition per n with >10 IDs per field, paired *t* test for DP intensity, repeated-measures one-way ANOVA with Tukey's multiple comparisons test for RhoA intensity). Red, orange, and yellow insets present magnifications of IDs in the cardiac tissue. **(C)** Lysates from Ctl or DP KD HL-1 cardiac myocytes were subjected to RhoA (17A) pull-down to isolate active GEFs, and lysates were probed for Ect2. Whole-cell lysates (WCLs) were probed for DP and Ect2, as well as GAPDH as a loading control. **(C′)** Quantification of blots demonstrated a significant decrease in active Ect2 in DP KD lysates (n = 3; paired *t* test, P = 0.003). Scale bars = 20 μm.

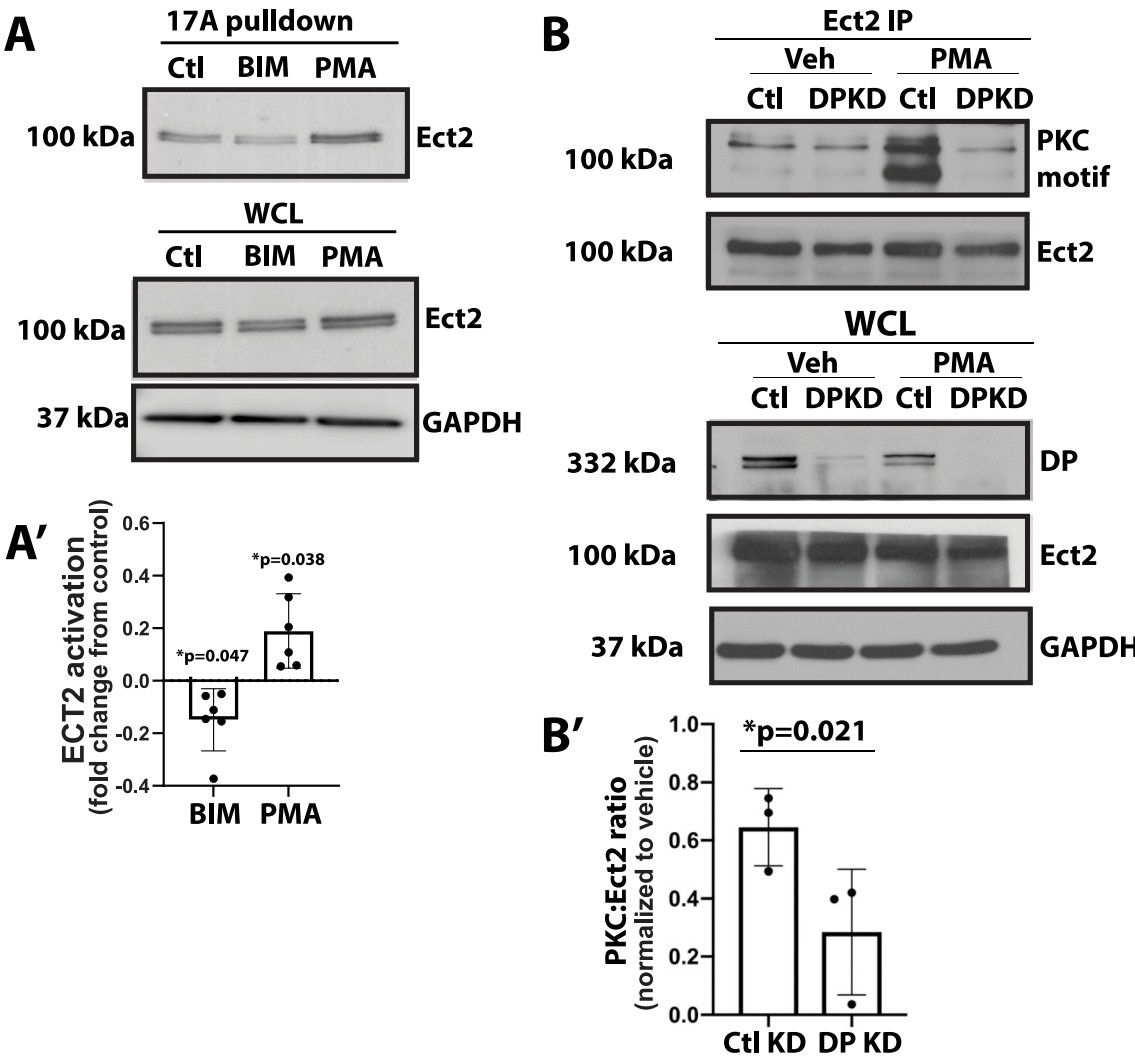

**Figure 5. DP regulates Ect2 activity via PKC-mediated phosphorylation.**
**(A)** Lysates from Ctl-, BIM-, or PMA-treated HL-1 cells were subjected to RhoA (17A) pull-down to isolate active GEFs and probed for Ect2. Whole-cell lysates (WCLs) were probed for Ect2, as well as GAPDH as a loading control. **(A′)** Quantification of blots demonstrated significantly more active Ect2 in cells in which PKC has been activated (PMA), compared with DMSO vehicle–treated cells, whereas BIM-treated cells in which PKC is inhibited show a decrease in Ect2 activity compared with DMSO vehicle–treated cells (n = 6; repeated-measures one-way ANOVA with Dunnett's multiple comparisons test, BIM, *$P$ = 0.047; PMA, *$P$ = 0.038). Data are from five independent experiments. **(B)** Ect2 was immunoprecipitated from Ctl or DP KD HL-1 cells treated with DMSO vehicle (Veh) or PMA-treated and blotted back with an antibody recognizing PKC-dependent phosphorylation or an antibody against Ect2. WCLs were probed for DP and Ect2, as well as GAPDH as a loading control. **(B′)** Quantification of the PKC motif:Ect2 ratio after normalization to vehicle-treated protein levels demonstrates significantly less PKC-dependent phosphorylation was detected on Ect2 in DP KD cells (n = 3; paired $t$ test, *$P$ = 0.021).

treated with either vehicle (DMSO) or PMA. This was followed by blot back with Ect2, as well as with an antibody that recognizes phosphorylated PKC substrates. Immunoblotting for phosphory-lated PKC substrates in vehicle-treated control KD NRVCMs showed a positive signal at the molecular weight corresponding to Ect2 (Fig 5B). Importantly, although control KD samples treated with PMA displayed a robust elevation of the PKC substrate motif signal compared with DMSO vehicle–treated cells, there was no apparent change in PMA-treated DP KD samples compared with the control (Fig 5B and B′). These findings indicate that DP is required for PKC-mediated phosphorylation of Ect2, possibly acting as a scaffold for these proteins.

## Ect2 junctional localization is disrupted in cardiac and epidermal cells from cardiocutaneous disease patients

Having delineated the role that DP plays in regulating the junc-tional localization and activity of Ect2 in cardiac models, we next asked whether the functional relationship between DP and Ect2 could have consequences for disorders associated with mutations in the *DSP* gene. Inherited mutations in *DSP* have been found to be a frequent causative factor underlying myocardial diseases including arrhythmogenic cardiomyopathy (AC), dilated cardiomyopathy, and myocarditis (27, 28, 29). Variants include Carvajal syndrome, a familial cardiocutaneous disorder defined

by the phenotypic presentation of woolly hair, palmoplantar keratoderma, and cardiac disease resulting from mutations in the *DSP* gene (30). Given the reported loss of DP localization to IDs of Carvajal patients, and our demonstration that DP was required for Ect2 localization at murine IDs (Fig 3), we asked whether Ect2 is also perturbed in tissue from a Carvajal patient (30). Using N-cadherin signal as a marker of IDs, we observed that ID localization of Ect2 is lost in cardiac sections from the Carvajal syndrome patient when compared to a healthy control tissue (Fig 6A).

A Carvajal disease patient reported in the literature harbored a single point mutation in the *DSP* gene that resulted in a premature stop codon and truncation of the third plakin repeat domain within the C terminus of DP (Fig 6B) (31). This patient presented with dilated cardiomyopathy, a form of cardiac disease characterized by an enlargement of the ventricular chamber and thinning of ventricular walls, along with cutaneous symptoms (30). Given the presentation of both cardiac and cutaneous phenotypes in this patient, we asked whether the junctional localization of Ect2 is impaired in keratinocytes isolated from this patient. To test this, control immortalized human keratinocytes (IHEKs) and keratinocytes isolated from the Carvajal syndrome patient (JD-1) were cultured for 2 d in high calcium medium to promote differentiation and desmosome maturation. Immunofluorescence analysis showed that DP was still able to localize to cell–cell junctions in these keratinocytes (Fig S3A). We also observed that Ect2 colocalized at keratinocyte cell junctions in IHEKs; however, the Ect2 junctional signal was significantly reduced in JD-1 cells (Fig S3A′).

To interrogate further the functional role that this DP mutant could be playing in the regulation of Ect2, we engineered the same point mutation that results in the truncated form of DP in the Carvajal syndrome patient into the DPII-GFP construct and inserted it into our adenoviral vector (Fig 6B). Interestingly, immunostaining of the Carvajal mutant DPII (DPII-Carvajal) in NRVCMs showed that this mutant form of DP still localized to cardiac cell junctions, similar to the observation in JD-1 cells (Fig 6C). This could suggest that loss of DP from Carvajal patient hearts could be due to the chronic effect of compromised cardiac function over time. Interestingly, immunostaining of Ect2 revealed a significant decrease of junctional Ect2 in DPII-Carvajal cells compared with WT DPII samples (Fig 6C and C′). To corroborate that the Carvajal mutation compromises the ability of DP to recruit Ect2, a GFP peptide trap was used to pull down the GFP-tagged WT DPII and DPII-Carvajal fusion proteins. Ect2 was pulled down along with WT DPII-GFP in the GFP-Trap, but was not detected in the GFP-only control or in the Carvajal-GFP pull-downs (Fig S3B). These data suggest that a portion of the DP C terminus is required for the localization of Ect2 to cardiac cell junctions, necessary for RhoA localization and junctional activity.

As loss of junctional RhoA in the absence of Ect2 could lead to functional consequences for cell–cell adhesive strength (32), we carried out a dispase assay on normal human epidermal keratinocyte (NHEK) monolayers subjected to Ect2 KD (Fig S3C and C′). A reduction in Ect2 resulted in increased fragmentation of confluent epithelial sheets, consistent with the importance of Ect2 in

maintaining tissue integrity through localization of active RhoA at desmosomes.

# Discussion

In this study, we identified the RhoGEF Ect2 as a novel component of the cardiac ID and keratinocyte desmosomes, which is required for the maintenance of an active pool of RhoA at IDs. Previous work has identified several RhoGEFs important for cardiac development and homeostasis (33, 34, 35), including sarcomeric-localized RhoGEFs such as p63RhoGEF (36) and obscurin (34). However, despite the established importance of subcellular positioning of RhoA activators, to our knowledge, this is the first report of a RhoA GEF that specifically localizes to ID structures.

Rho signaling through other molecular complexes present at the ID has been reported to activate downstream pathways through kinases and transcription factors including ROCK, YAP, MRTF-A, and PLCε, to regulate cardiac development and pathophysiology (20). For instance, the protein Myozap was identified as a component of the ID that directly binds to DP and the myosin phosphatase-RhoA interacting protein (MRIP), a negative regulator of Rho activity. Myozap activates MRTF/SRF-dependent signaling in a Rho-dependent fashion, and its loss has been linked to contractile dysfunction and cardiomyopathy (37). More recently, the GTPase Rnd1 was identified as a Myozap partner that promotes SRF signaling in a stretch-sensitive manner to regulate cell proliferation and hypertrophy (38). The AJ and ID protein α-catenin, which serves a cytoskeletal adapter role similar to DP in the AJ, regulates ID maturation and actomyosin contractility in a mechanosensitive fashion, which, in turn, controls the cytoplasmic-to-nuclear distribution of the transcription factor Yap to regulate cell proliferation (39).

The identification of a complex including both Ect2 and DP supports the existence of a more direct pathway to RhoA activation that could functionally complement mechanisms driven by Myozap. For instance, although Myozap activates long-term responses through SRF transcription, it is possible that more direct coupling of desmosomes to RhoA activation through Ect2 could lead to rapid changes in mechanical signaling. The desmosome–IF network plays a crucial role in balancing the forces of mechanical stress in many systems, allowing excess contractile activity to be offloaded (2). This capacity is particularly important in the ID, which facilitates force transmission and transduction between cells and across the myocardium (40). Indeed, using a desmoglein-2 (DSG-2) force sensor, high tensile loading was measured during contraction of human cardiac muscle cells, raising the possibility that desmosomes can directly sense and respond to mechanical stress (41). The DP-Ect2 module may serve as a sensor as a part of a relay that helps synchronize these signals across the tissue in a rapid fashion.

Here, we show not only that DP is present in a complex and colocalizes with Ect2, but also regulates Ect2 activity through PKC, a family of kinases that play critical roles in cardiac remodeling including during cardiac hypertrophy (42). We previously demonstrated that DP exists in a complex with PKCα through its

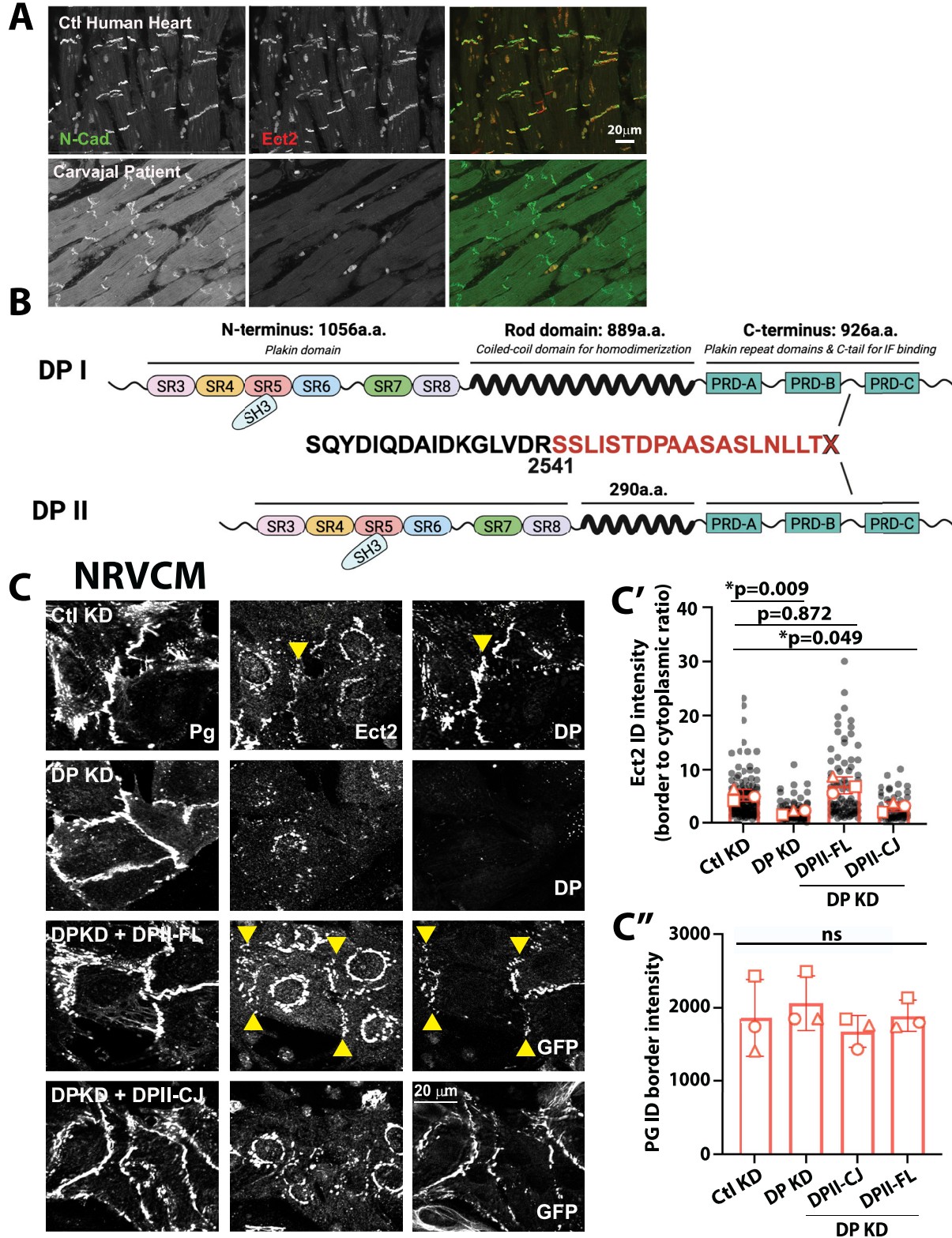

**Figure 6. Ect2 junctional localization is disrupted in cardiac tissues from Carvajal patients.**
**(A)** Sections from control (Ctl) or human Carvajal patient heart tissue were fixed and stained for Ect2, as well as N-cadherin to mark IDs. Ect2 staining at IDs was present in Ctl but not Carvajal samples. Scale bar = 20 μm. **(B)** Structure of DP from Carvajal patients, compared with WT DPI and DPII. Note that the map shows the location of the point mutation in DPI (aa 2541), whereas the engineered constructs are in DPII (aa 1941). **(C)** Isolated NRVCMs were treated with adenovirus encoding scrambled control (Ctl) or DP KD oligonucleotides and rescued with either WT DPII-FL or DPII-Carvajal constructs and stained for Ect2 and DP. Yellow arrowheads indicate junctions containing Ect2 and endogenous DP or the expression of the DPII rescue construct. Scale bar = 20 μm. **(C')** Quantification of fluorescence intensity of Ect2 at IDs showed

interactions with the armadillo protein plakophilin 2 (PKP2). Given that DP depends on PKP2 for its efficient localization to intercellular junctions, we expect that loss of PKP2, and possibly other desmosome proteins that could help stabilize the DP-PKP2 complex, would also interfere with association between Ect2 and active RhoA at IDs (26). Together, these observations suggest that desmosomes govern the localization and activity of a RhoA signaling module in the ID. The known dependence of PKC activity on the mechanical environment (43, 44) provides another mechanism to couple this module to changes in tissue mechanics during development and disease pathogenesis.

Several lines of evidence have linked aberrant Rho signaling to cardiac disease. In an animal model, a dominant active version of the downstream mediator ROCK was demonstrated to recapitulate fibrosis associated with heart failure in humans, where active ROCK1 accumulates (45). Consistent with these results, ROCK1−/− mice exhibited reduced fibrosis in the myocardium associated with reduced expression of extracellular matrix proteins and fibrogenic cytokines (46). On the other hand, up-regulation of ROCK activity as a result of EGFR inhibition was required for enhanced DP and DSG2 at area composita and increased junction length in HL-1 cardiomyocytes (47). Although we did not observe a difference in ID length or area in *Dsp* cKO IDs compared with WT control animals, a loss of attachment between adjacent cells and the breakdown of IDs was revealed by ultrastructural analysis of *Dsp* cKO cardiomyocytes (18). In addition to a role of dysregulated RhoA signaling in cardiac disease, Rho/ROCK signaling contributes to cell-fate decisions including cardiomyocyte identity through transcription of MRTF/SRF targets. Interfering with this pathway can prime cells to switch to an adipocyte lineage during cardiomyocyte differentiation (48).

We showed that Ect2's association with DP depends at least in part on a C-terminal region previously shown to comprise a portion of the IF binding domain, followed by 68 residues that dictate the strength of IF binding through post-translational modifications (49, 50, 51). This region is missing in a family with Carvajal disease, a syndrome associated with both cardiomyopathy and cutaneous symptoms including woolly hair (31). We observed a partial loss of Ect2 associated with DP at cell–cell junctions in keratinocytes isolated from the skin of a patient, and a loss of both DP and Ect2 at IDs in cardiac tissue from a Carvajal patient. This raises the possibility that a progressive loss of junctional Ect2 function occurs as the concealed phase transitions to more advanced cardiomyopathy state. In addition, a recent study in epithelial MDCK cells reported that IF association with desmosomes through DP is important for assembly of the myosin VI–E-cadherin mechanosensor that activates RhoA in epithelial cells, a function that was restored by introducing a minimal IF binding linker in which the DP C terminus was replaced with 111 amino acids of the related periplakin protein. Thus, it is plausible that loss of DP-IF interactions at the cardiac ID could result in decreased Rho through multiple mechanisms (52). Additional functions of

DP may also be lost during this transition, for instance, proteome degradation control through loss of the cardiac constitutive photomorphogenesis 9 (COP9) signalosome subunits 3 or 6 at the desmosome (53, 54).

Finally, we previously demonstrated that epithelial sheets generated from Carvajal keratinocytes have reduced ability to withstand mechanical stress, consistent with impaired adhesive strength. Here, we show silencing the RhoGEF Ect2 also results in reduced adhesive strength, consistent with the importance of DP-dependent recruitment of Ect2 and consequent localization of active RhoA at intercellular junctions for maintaining tissue integrity (Fig S3C and C') (55). Importantly, DP also associates with the RhoGAP ARHGAP32 in epithelial cells, suggesting that the desmosome scaffolds both positive and negative regulators of RhoA to tune contractile signaling (56). It will be interesting in the future to determine the extent to which DP-Ect-RhoA module works to convert mechanical cues into alterations in contractile signaling versus longer term transcriptional changes depending on actin remodeling, for instance, through the SRF/MRTF pathway, which has been demonstrated to be important for cardiac homeostasis and epidermal differentiation (48, 57, 58, 59, 60).

## Materials and Methods

### Isolation of NRVCMs, cell culture, mouse models, and human tissues

Neonatal rat ventricular cardiomyocytes were isolated as described from 1- to 3-d-old Sprague-Dawley rats (Charles River) (61). Briefly, hearts were dissected from neonatal rats, enzymatically digested, and resuspended in M199 medium (Lonza) containing 10% FBS and supplemented with 15 $\mu$M vitamin B12. To deplete isolates of more rapidly adhering cardiac fibroblasts, cells were plated for 2 h on plastic cell culture dishes. The remaining cell suspension, enriched for cardiac myocytes, was then filtered through a 40-$\mu$M filter, spun down, and resuspended in M199 medium containing 10% horse serum, 15 $\mu$M vitamin B12, and BrdU. Cells were then plated onto six-well dishes or coverslips that had been coated with collagen type IV. Medium was changed every 24–48 h with adenoviral infection being performed 48 h post-isolation. These protocols were conducted with the approval of the Northwestern University Institutional Animal Care and Use Committee, and all animal care protocols conform to National Institutes of Health guidelines and the recommendations of the Panel on Euthanasia of the American Veterinary Medical Association.

The HL-1 cardiac myocyte cell line was maintained with Claycomb Medium (Sigma-Aldrich) supplemented with 10% FBS (Atlanta Biologicals), 0.1 mM norepinephrine, 2 mM L-glutamine, and penicillin/streptomycin solution (Sigma-Aldrich). Cells were grown to confluency on plastic dishes precoated with a solution of

---

that although Ect2 was observed at junctions in DPII-FL–expressing cells, the Carvajal mutant was unable to restore Ect2 at IDs (n = 3; ROIs: 128 Ctl KD, 107 DP KD, 81 DP KD + DPII-FL rescue, 53 DP KD + DPII-CJ rescue IDs quantified, repeated-measures one-way ANOVA with Tukey's multiple comparisons test). **(C″)** Pg levels remained unchanged at IDs (n = 3; 5–10 fields per n, >10 ROIs per field; repeated-measures one-way ANOVA with Tukey's multiple comparisons test).

5 µg/ml fibronectin–0.02% gelatin. The Carvajal keratinocyte line JD-1 harbors DP with a deletion of G at position 7,622 within the coding sequence of DP (position 7,901 of GenBank/EMBL/DDBJ accession no. M77830). This mutation results in loss of the C-subdomain containing the plakin repeats and part of the upstream linker region and introduces 18 new amino acids downstream of the deletion.

JD-1 keratinocytes were isolated and immortalized with an HPV16 plasmid (pJ45216) that has early region genes driven by MoMLV-LTR as previously described (62). Normal adult keratinocytes immortalized with HPV16 E7 (IHEKs) were obtained from the Skin Biology and Diseases Resource-Based Center at Northwestern University Feinberg School of Medicine. Cells were cultured in DME/Ham's F-12 (3:1) supplemented with 10% FCS, 4 mM glutamine, 0.4 µg/ml hydrocortisone, 0.1 nM cholera toxin, 5 µg/ml insulin, and 10 ng/ml EGF.

Primary normal human epidermal keratinocytes (NHEKs) were isolated from human foreskin as previously described and grown in M154 medium supplemented with 0.07 mM $CaCl_2$, human keratinocyte growth supplement (HKGS), and gentamicin/amphotericin B solution (Thermo Fisher Scientific) (63). NHEKs were grown to confluency and switched to M154 medium supplemented with HKGS, gentamicin/amphotericin B, and 1.2 mM $CaCl_2$ for 24–48 h. Northwestern University Skin Biology and Disease Resource-Based Center (NU-SBDRC) maintains a protocol for collection and de-identification of human tissues (IRB protocol # STU00009443) and collection and handling of materials complied with ethical standards.

Mouse tissues were obtained from animals in which the *Dsp* gene was deleted in the mouse myocardium by breeding *Dsp*-floxed mice with heterozygous ventricular myosin light chain-2 Cre (MLC2v$^{(cre+)}$) mice (18). *Dsp* cKO mice and their control littermates were kept in a congenic C56BL/6 background. Hearts obtained from 8-wk-old mice were snap-frozen in liquid nitrogen and sectioned for immunofluorescence analysis.

Human tissues were obtained from Dr. Jeffrey Saffitz (Beth Israel Deaconess Medical Center, Boston, MA). De-identified control and Carvajal patient left ventricular myocardial tissue were obtained in paraffin-embedded blocks. Tissue sections were processed for immunohistochemistry as described below.

## DNA constructs, siRNAs, shRNAs, transfections, adenovirus production, and chemical reagents

The DPII-FL-GFP construct and adenovirus generation using Gateway recombination were previously described whereby DPII-GFP was cloned into the pAd CMV/V5-DEST vector or the LZRS retroviral backbone (Mitchell Denning; Loyola University Medical School) (17). The DPII-GFP Carvajal construct was created by generating a point mutation in the DPII-FL-GFP construct to match the reported mutation observed in the Carvajal patient reported in reference 31 (Epoch Life Science). The EmGFP BLOCK-iT PolIII miR RNAi Expression Vector kit (Thermo Fisher Scientific) was used to design and generate control and rat DP-specific oligonucleotides, which were cloned into the pAd CMV/V5-DEST vector using Gateway recombination (target sequences: 5'-AAACCGGAAACATCATCTCTT-3' and 5'-TGGTAATAGTTGACCCAGAAA-3'). Rat Ect2-specific oligonucleotide adenovirus was created in the same manner (5'-AGTAGG AGATGGTAACACACT-3' and 5'-TGAAGTGTCTGCCAAGCTAGT-3'). Adenovirus was generated using ViraPower Adenoviral Expression System (Thermo Fisher Scientific), and NRVCMs were infected for at least 90 min at 37°C with the virus in M199 medium. Retrovirus was generated using the Phoenix amphotropic cell line (Gary Nolan, Stanford University Medical School) maintained in DMEM (Thermo Fisher Scientific) with 10% FBS and antibiotics. Lipofectamine reagent was used to transfect Phoenix cell cultures with retroviral constructs, which were selected 48 h later with 1 mg/ml puromycin in culture medium. Retroviral supernatants were harvested from Phoenix cells cultured at 32°C and concentrated on a Centricon Plus-20 column (MilliporeSigma). Subconfluent NHEK cultures were incubated at 32°C in M154 (Thermo Fisher Scientific) along with 4 µg/ml polybrene (hexadimethrine bromide; Sigma Chemical Co.) and concentrated virus for 2–4 h. Transient transfections were carried out using DharmaFECT (GE Life Sciences) to introduce siRNA oligos (5'-UCAAAGUCCUGGAGCAAGA-3', 5'-GCAUCCAGCUUCAGACAA A-3', 5'-ACACCAAGAUCGCUCAGAA-3', 5'-GUGCAGAACUUGGUAAACA-3'; Thermo Fisher siGENOME) to silence DP in HL-1 cells. NHEKs were transfected using the Amaxa Nucleofector System according to the manufacturer's instructions. NHEKs were suspended in Ingenio Electroporation Solution (Mirus Bio) with siRNA for silencing Ect2 (Stealth RNAi siRNA HSS103051, HSS103052; Thermo Fisher Scientific) at a final concentration of 50 mM and electroporated using program X-001. Cells were plated at 60–70% confluence and incubated with DNA and DharmaFECT reagent premixed in serum-free medium. Fresh medium was added 24 h after transfection, and samples were lysed or fixed 48–72 h after transfection. The PKC activator phorbol 12-myristate 13-acetate (PMA) was diluted in DMSO and used at a concentration of 15 nM (Abcam); the PKC inhibitor bisindolylmaleimide (BIM) was diluted in DMSO and used at a concentration of 12.5 µM (MilliporeSigma) with treatment for 60 min.

## Antibodies

Primary antibodies used in this study are as follows: NW6 rabbit anti-DP directed against the C-terminal domain of DP (64), DP2.15 mouse antibody directed against DPI and DPII (Thermo Fisher Scientific), DP2.17 mouse antibody directed against DPI (Thermo Fisher Scientific), 115F mouse monoclonal antibody directed against DP (generous gift of David Garrod (65)) or purified from hybridoma culture (MilliporeSigma), 1,407 chicken anti-Pg (Aves Laboratories), mouse anti-N-cadherin (Invitrogen), sheep anti-N-cadherin (R&D Systems), mouse anti-α-catenin (1G5; Thermo Fisher Scientific), goat anti-α-catenin (LS Bio), mouse anti-Rho-GTP (New East Biosciences), rabbit anti-Ect2 (sc-1005; Santa Cruz Biotechnologies), mouse anti-Ect2 (LS Bio), rabbit anti-GAPDH (Sigma-Aldrich), rabbit anti-phospho-PKC substrate motif (Cell Signaling Technology), JL-8 mouse anti-GFP (Living Colors, Thermo Fisher Scientific). Peroxidase-conjugated anti-mouse or anti-rabbit secondary antibodies were used for Western blot analysis (Kierkegaard and Perry Laboratories, Inc.). Alexa Fluor 488/568/ 647–conjugated goat or donkey anti-mouse, anti-rabbit, and anti-chicken, anti-sheep, or anti-goat secondary antibodies were used for immunofluorescence assays (Thermo Fisher Scientific).

## Coimmunoprecipitation and Western blotting

For coimmunoprecipitation assays, cells were washed twice with PBS and lysed with an NP-40 buffer (100 mM NaCl, 10 mM Tris–HCl, pH 7.5, 0.2% NP-40, 10% glycerol) supplemented with cOmplete protease inhibitor cocktail (MilliporeSigma, Roche). After a 1-min vortex, the lysate was allowed to sit on ice for 10 min followed by centrifugation at 4°C for 30 min at 21,130$g$. 5% volume of the supernatant was saved as the input; the remaining supernatant was incubated with either rabbit-IgG primary antibody (negative control) and a rabbit anti-DP antibody (NW6) or mouse-IgG primary antibody (negative control). After rotation overnight with primary antibodies, protein A/G PLUS Agarose beads (Santa Cruz Biotechnology) were added to the samples and incubated for 1 h and then washed multiple times. IP samples were eluted using 3X Laemmli sample buffer (30% glycerol, 3% SDS, 188 mM Tris, pH 6.8, 5% $\beta$-mercaptoethanol) and boiled before proceeding to Western blot analysis. Affinity precipitation of exchange factors with the nucleotide-free RhoA mutant (G17A) was performed as described previously ([23], [66]). Briefly, cells were lysed in 20 mM Hepes, pH 7.6, 150 mM NaCl, 1% Triton X-100, and 5 mM MgCl$_2$, plus protease inhibitors. Equalized lysates were first precleared with 20 $\mu$g of purified GST alone (bound to glutathione-Sepharose beads) for 30 min at 4°C, followed by incubation with 20 $\mu$g of purified GST-RhoA (17A) for 60 min at 4°C. Purified samples were washed and eluted in Laemmli buffer with 5% $\beta$-mercaptoethanol, followed by running lysates on 7.5% SDS–PAGE gels followed by transfer to nitrocellulose for Western blot analysis, probing with primary and secondary antibodies directed against proteins of interest, DP and Ect2, with GAPDH as a loading control for whole-cell lysates.

## GFP-Trap

Retrovirally infected NHEKs were cultured in 1.2 mM CaCl$_2$-containing media for 48 h and lysed in NP-40 buffer (10 mM Tris–HCl, pH 8, 100 mM NaCl, 0.2% NP-40, 10% glycerol, with protease inhibitor tablet), and the ChromoTek GFP-Trap experiment was performed following the manufacturer's instructions (Proteintech).

## Proximity ligation assay (PLA)

For the Duolink proximity ligation assay (PLA) protocol, the immunofluorescence protocol described for 2D was followed through the addition of the primary antibody. After primary antibody incubation, the PLA protocol for 2D and tissue samples was performed according to the manufacturer's instructions with mouse and rabbit plus and minus probes (Sigma-Aldrich). Directly after the PLA procedure was completed, plakoglobin immunofluorescence staining was performed with 1-h primary and secondary antibody incubations to label cell–cell borders. A DP knockdown was included as a control for positive DP-Ect2 PLA signal in NRVCMs. Additional information on troubleshooting, image acquisition, and analysis is detailed in reference [67]. Samples were mounted with Duolink In Situ Mounting Medium containing DAPI

(Sigma-Aldrich), and images were acquired using an AxioVision Z1 system (Carl Zeiss) with Apotome slide module, an AxioCam MRm digital camera, and a 40× (1.4 NA; Plan-Apochromat oil objective). ImageJ software was used to quantify the percent area of the PLA signal per field.

## Immunofluorescence (cultured cells)

NRVCMs, JD-1s, and NHEKs grown on glass coverslips were washed several times with PBS, then fixed using ice-cold methanol for 2 min at −20°C or by using 4% PFA for 10 min at RT, and then permeabilized with 0.2% Triton X-100 in PBS for 10 min at RT. For Rho-GTP staining, fixation was performed with ice-cold 10% trichloroacetic acid (TCA) in water for 15 min followed by a 20-min extraction in ice-cold 0.2% Triton X-100. After fixation, cells were then washed several times with PBS before incubating cells in blocking buffer (1% BSA, 2% normal goat serum in PBS) for 30–60 min at RT before antibody incubations. Primary antibodies diluted in blocking buffer were added to coverslips and incubated at 37°C for 1 h or overnight at 4°C followed by multiple washes in PBS. Alexa Fluor secondary antibodies diluted in blocking buffer (Thermo Fisher Scientific) were then added and incubated for 30–60 min, followed by a 2-min 2 $\mu$g/ml incubation with DAPI and then several PBS washes and mounting of coverslips in ProLong Gold (Thermo Fisher Scientific). All images were acquired using an AxioVision Z1 system (Carl Zeiss) with Apotome slide module, an AxioCam MRm digital camera, and either a 20× (0.8 NA; Plan-Apochromat), 40× (1.4 NA; Plan-Apochromat oil objective), or 100× (1.4 NA; Plan-Apochromat), and brightness and contrast and image cropping were applied uniformly within each experiment. Manipulations were performed using Zen microscopy software (Carl Zeiss) or Adobe Photoshop or Adobe Illustrator except for Figs 1A and S1A where images were acquired using N-SIM Super-Resolution Microscope (Nikon N-SIM; Nikon) using an oil-immersion objective lens (CFI SR Apochromat 100×, 1.49 NA; Nikon) for 3D-SIM. Nine optical sections were taken at a step size of 0.2 μm. Nikon Elements Advanced Research with an N-SIM module was used to reconstruct the structured illumination images. Illumination contrast modification, high-resolution noise suppression, and out-of-focus blur suppression were set with values for image reconstruction as in the table below.

**N-SIM image reconstruction values.**

|                                    | 488  | 561  | 640  | 405  |
|------------------------------------|------|------|------|------|
| Illumination contrast modification | 0.51 | 1.02 | 0.50 | 1.00 |
| High-frequency noise suppression   | 1.13 | 0.56 | 1.00 | 0.56 |
| Out-of-focus blur suppression      | 0.18 | 0.17 | 0.1  | 0.11 |

## Immunofluorescence (tissues)

Tissue samples were fixed with 10% neutral-buffered formalin, embedded in paraffin, and cut into 4- to 5-$\mu$m sections. Paraffin-embedded sections were baked at 60°C overnight and deparaffinized using xylene. Samples were run through a series of

ethanol and PBS dips, and slides were permeabilized in 0.5% Triton X-100 in PBS. Antigen retrieval for paraffin-embedded sections was performed by heating samples to 95°C in 0.01 M citrate buffer. Sections were blocked in blocking buffer (1% BSA, 2% normal goat serum in PBS) for 1 h at 37°C and incubated with primary antibodies overnight at 4°C in a humid chamber and secondary antibodies at 37°C for 1 h followed by mounting using ProLong Gold (Thermo Fisher Scientific). All the images were acquired using an AxioVision Z1 system (Carl Zeiss) with Apotome slide module, an AxioCam MRm digital camera, and either a 20× (0.8 NA; Plan-Apochromat), 40× (1.4 NA; Plan-Apochromat oil objective), or 100× (1.4 NA; Plan-Apochromat).

### Dispase assay

IHEK and JD-1 keratinocytes were cultured on 60-mm six-well dishes, and confluent monolayers were incubated in 2 ml of dispase (2.4 U/ml; Roche Diagnostics GmbH) for more than 30 min. Released monolayers retained in the six-well dishes were placed on a rotary shaker, and fragments were counted and images collected using a Leica MZ6 dissecting microscope equipped with a Leica DFC295 digital camera using Leica Application Suite software (55).

### Quantification methods and statistics

All quantifications are presented as the mean ± SD. Repeated-measures one-way ANOVA with Tukey's multiple comparisons or Dunnett's multiple comparisons or paired or unpaired $t$ tests were used for analyses as appropriate, and the specific analyses performed are included in each figure legend. $F$ tests or Brown–Forsythe tests comparing variances calculated from the means of the three biological replicates were performed and showed no significant differences in variance between groups. Statistical tests were performed on the means of ROIs from three or more independent experiments. $P < 0.05$ was considered significant. Junctional fluorescence intensity measurements shown in Figs 4B′ and C″ and S2A–D were calculated using the following method. Border segmentation was performed on the plakoglobin channel using MaxEntropy method autothresholding. With segmented areas as ROIs, mean intensity was measured in each channel. For figures showing intensity data for high and low DP intensity regions, high DP junctions are those with the top 20% DP intensity values for each given condition; low DP junctions represent the bottom 80%. Given the high number of ROIs obtained using this thresholding method, we report data as the means of total intensity. For all other figures, junctional ROIs were selected manually by segmenting on junctional Pg signal, averaging pixel intensities within the ID and an area of equal size adjacent to the ID (cytoplasm), and each data point was plotted as a ratio of these averages. For statistical analysis presented as fold change (bar graphs), the mean of data from control samples was assigned a value of 1 and all other samples were calculated relative to the control. All densitometry quantification was normalized to respective loading controls.

## Data Availability

All data are available in the main text or supplemental data, and raw data or other unique resources are available upon request to the authors.

## Supplementary Information

## Acknowledgements

The authors would like to acknowledge Jeffrey Saffitz for contributing patient sections and Mariana De Niz in the Center for Advanced Microscopy for assistance with SIM images. This research utilized resources and instrumentation provided by Northwestern University core facilities: Skin Biology and Diseases Resource-based Center (P30AR075049) and Center for Advanced Microscopy (RRID: SCR_020996) CCSG P30 CA060553. The work was supported by the following grants: NIH R01 AR041836, R01 AR043380, and R01 CA228196, and the JL Koetsier Mayberry Endowment to KJ Green; American Heart Association Fellowships to CY Kam and H Zarkoob; and NIH R01HL142251, NIHR01 HL162369, and LEXEO Therapeutics Inc. to F Sheikh. F Sheikh was a cofounder of Stelios Therapeutics Inc. (acquired by LEXEO Therapeutics Inc.) and is a cofounder and shareholder of Papillon Therapeutics Inc. and MyoTherapeutix Inc., as well as is a consultant and shareholder of LEXEO Therapeutics Inc.

### Author Contributions

H Zarkoob: conceptualization, data curation, formal analysis, validation, investigation, visualization, and methodology.
CY Kam: conceptualization, data curation, validation, investigation, methodology, and writing—original draft.
JL Koetsier: validation, investigation, and methodology.
E McCarthy: data curation, software, formal analysis, and visualization.
A Jaiganesh: validation, investigation, and methodology.
DP Kelsell: resources.
F Sheikh: resources.
LM Godsel: formal analysis, investigation, visualization, project administration, and writing—review and editing.
KJ Green: conceptualization, resources, supervision, funding acquisition, methodology, project administration, and writing—original draft, review, and editing.

### Conflict of Interest Statement

The authors declare that they have no conflict of interest.

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
