## [Reviewer comments · Life Science Alliance]

RhoGEF Ect2 Supports RhoA activity at Cell-Cell Junctions through Desmoplakin

Hoda Zarkoob, Chen Kam, Jennifer Koetsier, Erin McCarthy, Avinash Jaiganesh, David Kelsell, Farah Sheikh, Lisa Godsel, and Kathleen Green

DOI: <https://doi.org/10.26508/lsa.202503454>

Corresponding author(s): Kathleen Green, Northwestern University and Lisa Godsel, Northwestern University Feinberg School of Medicine

Review Timeline:

Submission Date:	2025-07-16
Editorial Decision:	2025-08-12
Revision Received:	2026-01-29
Editorial Decision:	2026-02-17
Revision Received:	2026-02-27
Accepted:	2026-03-02

Scientific Editor: Tim Fessenden

Transaction Report:

August 12, 2025

Re: Life Science Alliance manuscript #LSA-2025-03454

Dr. Kathleen J Green
Northwestern University
Pathology
303 E. Chicago Ave
303 E Chicago Ave
Chicago, Illinois 60611

Dear Dr. Green,

Thank you for submitting your manuscript entitled "RhoGEF Ect2:Desmoplakin Association at Intercellular Junctions: Implications for Carvajal Disease" to Life Science Alliance. The manuscript was assessed by expert reviewers, whose comments are appended to this letter.

As you will see, all reviewers appreciated the observations on a Rho GEF acting locally at desmosomes and intercalated discs in a manner that depends on Desmoplakin. Reviewers overall requested greater clarity on methods, details on mechanism of Ect2 recruitment, and evidence for Ect2 interactions in different cell types. Reviewers 1 and 4 sought validation of the Ect2-DP interaction shown in Fig 1, as well as important details on imaging, quantification, and statistics. Reviewers 1, 2, and 3 made related suggestions requesting biochemical evidence for Ect2-DP interaction in different cell types. A suitable revision should include data to resolve these requests. Next, Reviewer 3 asked if DSG2 and PKP2 may participate in local Ect2 recruitment, and Reviewer 4 requested closer examination of effects on PG, and we agree these issues should be resolved in a manner of your choosing. While a revised manuscript must respond to all reviewer comments in some manner, the remaining points are left to your discretion.

Thank you for this interesting contribution to Life Science Alliance. We are looking forward to receiving your revised manuscript.

Sincerely,

B. MANUSCRIPT ORGANIZATION AND FORMATTING:

Reviewer #1 (Comments to the Authors (Required)):

This new manuscript from Zarkoob, Kim, and colleagues describes their efforts on the relationship between Desmoplakin (DP) and RhoGEF Ect2 at cardiomyocyte cell-cell junctions. The work builds on studies from the Green lab into desmosome function in cardiac tissue, especially the role of individual desmosome proteins in signaling. Previously, they discovered that disrupting the connection between intermediate filaments (IFs) and DP reduced RhoA recruitment to and activation at cell-cell junctions. The RhoGEF Ect2 is an important regulator of RhoA, so they explored its localization and function at cardiomyocyte cell-cell junctions. They found that: 1) Ect2 localized to newly established cell-cell contacts in cultured rat neonatal cardiomyocytes (NRVCMs) and the intercalated disc (ICD) in mouse heart; 2) Ect2 associated with/was proximal to DP in co-IP and PLA experiments; 3) loss of DP reduced Ect2 recruitment to cell-cell contacts in NRVCMs (shRNA knockdown) and in the mouse heart (conditional knockout of Dsp); 4) depletion of DP and Ect2 reduced Rho GTPase activity in NRVCMs, HL-1 cells, and mouse heart ICD; 5) PKC phosphorylation regulates Ect2 activity via DP; and 6) Ect2 localization to cell-cell contacts is disrupted in tissue samples from a Carvajal Syndrome patient.

Overall, this is a well-executed, clearly written, and complete study that provides new and important details on Ect2 and DP function in cardiac cells. The data support all major points and conclusions. I have one major comment and several minor comments, listed below:

Major criticism:

Critical details about image acquisition and analysis are lacking from the methods. For example, was SIM used for all NRVCM imaging? If so, was a single section or a max projection used for image analysis? How was the border/cytoplasm analysis done? How was the border designated in these experiments? What was the n value and # of biological replicates for each analysis?

Minor criticisms:

Using a combination of cell lines, primary cultures, and mouse and human cardiac tissues is a strength in the study and should be commended. That said, the HL-1 cardiac myocyte cell line is the weakest of the systems, given its morphological and physiological quirks. I understand why the authors used this system for their pulldown and IP experiments, but where do DP, Ect2, and active Rho localize in these cells? This data would strengthen the HL-1 results.

The authors describe Ect2 as a "binding partner" of DP. While they never explicitly state that DP binds directly to Ect2, I feel that "binding partner" implies a direct interaction. Direct binding is not tested in this study. I recommend that the authors use another term or state clearly that the binding partner does not imply a direct interaction.

The Ect2 perinuclear staining is interesting (or odd?). While the authors suggest that the IF signal may be nonspecific, it appears to decrease in the Ect2 and DP KD cells. Is there evidence for nonspecific binding by western (i.e., another prominent band)? Given the recent evidence for desmosome-ER interactions (Bharathan et al., 2023), I wouldn't be quick to dismiss this staining pattern. It could reflect some interesting biology!

Fig1 B': the linescans are not informative and the difference in colocalization is not readily apparent. At a minimum, Pearson's correlation coefficient analysis should be used to quantify and statistically compare colocalization.

Fig1. C': Why not compare the average of all three points? Also, how were the stats done?

In Fig 2A, Fig 4A, and Fig6 C, I dislike how the PG and DAPI signals were combined in blue and overlaid with the grayscale Ect2 signal. I found it confusing. Why merge the DAPI with PG? Why blue rather than making Ect2 green and PG red (or better, magenta) and showing the merge (minus DAPI)? Also, why is DP green rather than greyscale? The green color is unnecessary.

Fig 4B": it would be helpful to show examples of both "high" and "low" DP at the ICD.

Fig 6B: a cartoon schematic of DP11 would be helpful.

Reviewer #2 (Comments to the Authors (Required)):

This paper describes the role of the desmosomal protein desmoplakin in recruiting the RhoA activator (GEF) Ect2 to intercalated disks (IDs) of cardiac myocytes and junctions of cultured cardiac-derived cells. In addition, a mechanism of activation of Ect2 by aPKC is proposed, and a significant link to human pathology illustrated by examining epidermal cells isolated from patient with a cardio-cutaneous disease (Carvajal syndrome). This paper advances significantly the field, since little attention had been devoted so far to the localisation of Rho family regulators at IDs, and mechanistic insights into disease pathogenesis are rare. The data are strongly supportive for all the data of the papers which contain appropriate controls, multiple approaches, and an impeccable description.

This Reviewer suggests a few additional experiments that could nicely complement the data:

1. IP experiments were only carried out using human keratinocytes. Was a coIP experiment attempted using cardiac tissue or cells? This would strengthen the biochemical argument for interaction. A biochemical experiment of direct interaction is also missing (see below).
2. Since the authors observe a decrease of DP junctional labelling in cells deleted of Ect2, does inhibition of RhoA activity perturb DP localisation at junctions of cardiac-derived cells in vitro?
3. The authors convincingly show that RhoA activity is altered in cells depleted of Ect2, but it would be interesting to provide additional functional assays, for example measuring strength of adhesion, cell contractility etc.
4. The authors say that Ect2 labelling is "completely lost" in cells from Carvajal syndrome patients (Fig. 6A). However some labelling is still detectable. What structures is Ect2 associated with in these cells?
5. The authors use a DP mutant to infer about the region of DP implicated in binding to Ect2. If direct binding occurs, the binding and the mapping of the binding site can be easily demonstrated with pulldown assays using purified bait and preys.
6. A pulldown or other experiments could also be useful to compare affinities of binding of Ect2 to either DP or the cadherin-catenin complex.
7. The Dsp cKO myocytes (Fig. 3B) show junctions that are much more elongated compared to WT. Has electron microscopy been done in these cells to understand better the ultrastructural/morphological effects of the conditional KO?

The text is very clearly written, the statistics is good, and the data are presented in a logical manner, which flows beautifully.

Minor points

- calcium medium, not media (medium is singular, media is plural in latin).

Reviewer #3 (Comments to the Authors (Required)):

Zarkoob et al. investigated the functional interaction between the RhoGEF Ect2 and the desmosomal component desmoplakin (Dsp), focusing on the intercalated disc (ID) and linking their findings to Carvajal disease. The authors demonstrated the interdependence of Ect2 and Dsp, with Rho as a regulatory component and involvement of PKC.

Overall, this study provides important new insights into the composition and function of ID components and is of interest to both the cell biology and cardiology fields. To fully realize the potential of this study, we suggest the following:

Major Comments:

1. Specificity of the effects to desmoplakin:

The authors propose Ect2 function and localization to be dependent on Dsp, but how specific are the observed effects to Dsp, as opposed to general desmosomal disruption, given that Dsp is the major stabilizing component of desmosomes? The authors primarily used plakoglobin (Pg) for counterstaining; however, Pg is also a component of adherens junctions (AJs). To distinguish the role of Dsp from desmosomal components, it would be important to assess the dependence of Ect2 localization and function on other desmosomal proteins such as desmoglein-2 (Dsg2) or plakophilin-2 (Pkp2).

2. Functional Link to Carvajal Mutation:

The authors associate the localization of Ect2 at the ID with the Carvajal-associated Dsp mutation. Is Rho activity as a consequence then reduced by the Dsp mutation? Additionally, is Dsp still detectable at the ID in patient samples?

3. Ect2 Reduction upon Dsp Knockdown:

The authors report reduced localization of Ect2 at the ID following Dsp knockdown. Is this due to impaired recruitment of Ect2 to the ID, or does Dsp influence Ect2 expression (at the RNA/protein level) or its stability/turnover?

Furthermore, KD of Ect2 also reduces the relative intensity of Ect2 at the ID. Would one not expect a uniform reduction in such a case?

4. Immunoprecipitation Analysis (Supp 1D):

The immunoprecipitation experiment is a key approach to confirm the structural association between Dsp and Ect2, supporting the data from the PLA assay. However, it was performed only in keratinocytes. To substantiate the relevance of this interaction at the ID-the central focus of the study-please provide a corresponding IP analysis in cardiomyocytes. E.g. in Fig 5B: Was Dp pulled-down with Ect2?

Minor Comments:

Fig 1C: PLA signal appears not only at cell-cell junctions but also in the cytoplasm/nucleus-can the authors clarify this observation? Is this expected? Are appropriate controls for the PLA assay included (e.g., single antibody or isotype controls)?

Fig 2A: The authors state that DP is reduced upon Ect2 knockdown. Please provide quantification of DP staining to support this claim.

Fig 3C: Please provide images of cardiomyocytes in a longitudinal orientation to better visualize the intercalated discs.

Fig 5A: A loading control for total protein after pulldown is needed to ensure equal input across conditions.

Fig 5A': The p-value indication is unclear: is the comparison made between BIM and PMA or BIM and control? What exactly does the y-axis represent? Please define the unit or measurement.

Supp Fig 2: The y-axis label is missing-please specify what is being measured. Where were the regions of interest (ROIs) placed? Clarify how localization was determined.

Fig 1B': Axis labels are missing from the graph. Indicate the position of the line scan in the corresponding 2D image for spatial reference.

Fig 5B': Which western blot does this quantification correspond to? Is it for vehicle-treated or PMA-treated samples? Clarify what is meant by "PKC:Ect2" - does this refer to a PKC phosphorylation motif?

Statistical Analysis: For the statistical tests applied, assessment of normal distribution and homogeneity of variance is required. Please indicate whether these assumptions were tested and met. For each experiment, provide the number of independent biological and technical replicates performed.

Reviewer #4 (Comments to the Authors (Required)):

The manuscript presents a novel and intriguing finding that the RhoGEF Ect2 localizes to cardiac intercalated discs (IDs) and keratinocyte desmosomes, contributing to the maintenance of an active RhoA pool at these junctions. The work addresses a significant gap in the current understanding of spatial regulation of RhoA signaling in cardiac tissue. The identification of Ect2 as the first RhoA GEF reported to localize to ID structures specifically is potentially impactful for both cardiac biology and broader cell-cell junction research. But this reviewer has few concerns regarding the experimental parts of the manuscript.

Reviewer comments:

1. In supplementary fig 1A, the images are not aligned.

2. In fig 1 B', what does the axes represent? There is no clarity on how many experimental repeats were done. To demonstrate that Ect2 displays strong co-localization with DP, please quantify the co-localization of Dp with Ect2 and N-cad with Ect2.

3. In Fig. 2A, the authors describe Pg as unaltered after Dp KD, consistent with their previous study. However, the provided images do not show equal intensity of Pg between Ctrl KD and DP KD. The authors should either provide better images between CTI and DP KD or quantify the PG and show that it was unaltered between Ctrl and DP KD. Because if PG is altered one could not claim that DP is responsible for ect2 localization.

4. As explained above for Figure 2a, please provide better images for PG in Fig 4a, to demonstrate that PG is not altered in DP KD and to quantify its intensity at the IDs, thereby showing that it remains unaltered.

5. Please provide a detailed explanation of the quantification methods used (e.g., cell borders/IDs, including the number of cell borders/images/IDs quantified) for each quantification performed in this manuscript). The provided information is insufficient for repetitions.
6. In Fig. 6c, please also quantify PG in all conditions or provide more explicit images that show no alterations in PG. For the reviewer, it appears that, at least in the provided images, the DPKD+DP11-FL Ect2 signal colocalized more with PG than with DP.
7. The authors should also discuss how the observed interaction of DP-Ect2 is essential for desmosome function in cardiomyocytes, since available literature suggests DP is necessary for ROCK-mediated desmosome assembly in cardiomyocytes (PMID: 36795511)

Dr. Tim Fessenden
Scientific Editor
Life Science Alliance

Dear Dr. Fessenden,

Thank you for your enthusiasm for our work and willingness to consider our revised manuscript #LSA-2025-0345, with revised title “RhoGEF Ect2 Supports RhoA Activity at Cell-Cell Junctions through Desmoplakin” for publication in *Life Science Alliance*. We found the reviewers' comments to be very helpful in improving our manuscript and thank them for their insights.

We are submitting a revised manuscript (revisions highlighted in yellow) addressing the comments of the reviewers as well as a general response to your editor summary, a list of changes to the figures, and a point-by-point response to reviewers' comments. All authors have agreed to the submission of this revised manuscript. Thank you for considering our revised manuscript for publication in *Life Science Alliance*.

Sincerely,

Lisa M. Godsel, Ph.D.

Kathleen J. Green, PhD
Joseph L. Mayberry Sr. Professor
Departments of Pathology and Dermatology
Northwestern University
303 E Chicago Ave.
Chicago, IL 60611
312-503-5300
kgreen@northwestern.edu

Editor Comments

“As you will see, all reviewers appreciated the observations on a Rho GEF acting locally at desmosomes and intercalated discs in a manner that depends on Desmoplakin. Reviewers overall requested greater clarity on methods, details on mechanism of Ect2 recruitment, and evidence for Ect2 interactions in different cell types. Reviewers 1 and 4 sought validation of the Ect2-DP interaction shown in Fig 1, as well as important details on imaging, quantification, and statistics. Reviewers 1, 2, and 3 made related suggestions requesting biochemical evidence for Ect2-DP interaction in different cell types. A suitable revision should include data to resolve these requests. Next, Reviewer 3 asked if DSG2 and PKP2 may participate in local Ect2 recruitment, and Reviewer 4 requested closer examination of effects on Pg, and we agree these issues should be resolved in a manner of your choosing. While a revised manuscript must respond to all reviewer comments in some manner, the remaining points are left to your discretion.”

Authors response to Editors' Comments in Summary (above)

As detailed in the responses included below, we addressed issues raised by the editor and referees in a substantially revised manuscript (changes highlighted in yellow) and a list of changes to figures below. We included more details on the methods used for imaging, quantification and statistics. All graphs now include the averages from each “N” of the experiment used for statistical analysis. Supplementary figure 1D now includes the results from a coimmunoprecipitation of DP and Ect2 in the cardiac line, HL-1s to complement the data from keratinocytes (Supplementary figure 1E). The signal, while low, is specific. Furthermore, we have included a functional assay in Supplementary Figure 3C, C' demonstrating the importance of Ect2 expression for maintaining robust cell-cell adhesion. We also discuss results from a collagen contraction assay, with data included in the letter for the referees' information. We opted not to include these data in the paper due to challenges associated with variability in the rescue arms of the assay. We modified figures that included staining for Pg by separating the channels to better illustrate the localization of Pg in fluorescence images. Furthermore, we have included quantification for Pg and DP to illustrate that Pg was minimally affected by Ect2 silencing (Fig. 6C"; Supplementary Figs 2B,D). While we did not experimentally extend our study to PKP2 and Dsg2, we discuss potential contributions to Ect2/RhoA regulation from previous work, including our own work demonstrating the dependence of DP and RhoA on PKP2. We have revised the text to make clear the limitations of the study, to exert caution in the nature of the interaction between DP and Ect2, also reflected by a change in the title to “RhoGEF Ect2 Supports RhoA Activity at Cell-Cell Junctions through Desmoplakin: Implications for Carvajal Disease.” When considered collectively, we firmly believe that our data strongly support Ect2 as a functionally important RhoGEF that contributes to RhoA localization and activity at desmosomes and intercalated discs, in a DP-dependent fashion.

A comprehensive list of changes to the figures is provided here, followed by a point-by-point response to the referees.

Changes to the figures by figure:

Figure 1:

Fig.1B' : DP, Ect2 and N-cad line scans have been removed

Fig. 1C: Pg label has been changed to red to separate it from the DAPI channel and better illustrate cell borders.

Fig.1C' has been changed to show averages for the PLA experiments; each paired "N" is indicated with a different shape.

Figure 2:

Fig.2A,B: Removed the Pg/DAPI/Ect2 blue overlay and made DP black and white. Arrows have been added to highlight the NR/VCM cell-cell borders in 2A indicating colocalization of DP and Ect2.

Figure 3:

Fig. 3A: Added boxes showing examples of Ect2 staining at Pg-positive IDs with high DP or minimal DP.

Fig B,C: Added boxes to highlight alpha catenin and N-cadherin positive IDs with reduced Ect2. Ect2 border intensities at DP "high" vs "low" junctions in the DP cKO is shown in Supplementary Figure 2A.

Figure 4:

Fig.4B: Added an inset for a DP positive region in the Dsp cKO (orange box) as an example of a medium level of DP at the ID. Collectively these examples show that RhoA tracks with DP levels.

Pg and DP border intensities for 4A are in Supplementary Fig. 2B, B', and Pg intensity is in D.

Figure 5:

Fig 5A' and B': Added fold change and normalization to vehicle to the Y axis for each, respectively.

Figure 6:

Fig 6B: Added a DP II to the schematic currently there for DP I.

Fig 6C": Added Pg ID border intensity showing no change from CTL with DPKD or the rescue with DP II or DP CJ.

Fig 6C: Removed the Pg overlay with Ect2 and made the DP B&W.

Suppl Fig 1:

Fig1A: Fixed the alignment of the figure panels

Fig 1D: Added HL-1 pull down to show Ect2 in DP immunoprecipitates

Suppl Fig 2:

Suppl Fig2A: Ect2 border intensities and "high" and "low" DP borders in the Dsp cKO

Suppl Fig2B, B': Pg and DP border intensities corresponding to 4A

Suppl. Fig2D: Pg intensity at IDs corresponding with Fig. 4B showing no significant change.

Suppl Fig 3:

Suppl 3C, C': Added a functional disperse assay showing that siEct2 treatment results in significant loss of epithelial sheet integrity.

Reviewer #1 (Comments to the Authors (Required)):

Reviewer Comments: *This new manuscript from Zarkoob, Kim, and colleagues describes their efforts on the relationship between Desmoplakin (DP) and RhoGEF Ect2 at cardiomyocyte cell-*

cell junctions. The work builds on studies from the Green lab into desmosome function in cardiac tissue, especially the role of individual desmosome proteins in signaling. Previously, they discovered that disrupting the connection between intermediate filaments (Ifs) and DP reduced RhoA recruitment to and activation at cell-cell junctions. The RhoGEF Ect2 is an important regulator of RhoA, so they explored its localization and function at cardiomyocyte cell-cell junctions. They found that: 1) Ect2 localized to newly established cell-cell contacts in cultured rat neonatal cardiomyocytes (NRVCMs) and the intercalated disc (ICD) in mouse heart; 2) Ect2 associated with/was proximal to DP in co-IP and PLA experiments; 3) loss of DP reduced Ect2 recruitment to cell-cell contacts in NRVCMs (shRNA knockdown) and in the mouse heart (conditional knockout of *Dsp*); 4) depletion of DP and Ect2 reduced Rho GTPase activity in NRVCMs, HL-1 cells, and mouse heart ICD; 5) PKC phosphorylation regulates Ect2 activity via DP; and 6) Ect2 localization to cell-cell contacts is disrupted in tissue samples from a Carvajal Syndrome patient.

Overall, this is a well-executed, clearly written, and complete study that provides new and important details on Ect2 and DP function in cardiac cells. The data support all major points and conclusions. I have one major comment and several minor comments, listed below:

Response: We thank the reviewer for their positive statements and helpful suggestions that are included below along with our responses.

Reviewer comments

Major criticism: Critical details about image acquisition and analysis are lacking from the methods. For example, was SIM used for all NRVCM imaging? If so, was a single section or a max projection used for image analysis? How was the border/cytoplasm analysis done? How was the border designated in these experiments? What was the n value and # of biological replicates for each analysis?

Response: We have included detailed information on imaging, how border/junctional intensity measurements were performed, and the number of biological repeats and individual data points, in the materials and methods and/or figure legends as requested.

Reviewer comments:

Minor criticisms: Using a combination of cell lines, primary cultures, and mouse and human cardiac tissues is a strength in the study and should be commended. That said, the HL-1 cardiac myocyte cell line is the weakest of the systems, given its morphological and physiological quirks. I understand why the authors used this system for their pulldown and IP experiments, but where do DP, Ect2, and active Rho localize in these cells? This data would strengthen the HL-1 results.

Response: We agree that the HL-1 cell line has limitations, including exhibiting ill-defined, disorganized intercellular junctions. Our ability to interrogate these junctions was further limited by the fact that a Santa Cruz Ect2 antibody with broad utility for both biochemical and microscopical analysis across multiple cell types, is no longer available. Despite efforts to make our own antibody against the same antigen (the company we were working went out of business while in the process of generating the reagent) and to identify other sources of Ect2 Abs, we haven't identified a comparable reagent that is as sensitive. Given this challenge in obtaining high quality staining in this line, we have used HL-1 cells for biochemical analysis only. In addition to the data in the original submission, we now include new data showing the presence of DP and Ect2 in co-immunoprecipitates, consistent with these proteins being in proximity. The signal, while low, is specific. Altogether our data indicate that DP and Ect2 are in close proximity.

Reviewer comments: *The authors describe Ect2 as a "binding partner" of DP. While they never explicitly state that DP binds directly to Ect2, I feel that "binding partner" implies a direct interaction. Direct binding is not tested in this study. I recommend that the authors use another term or state clearly that the binding partner does not imply a direct interaction.*

Response: We appreciate the reviewer's concern, and we have revised the manuscript so that a direct interaction is not implied by our description in the text. We have also changed the title to the more conservative: "RhoGEF Ect2 Supports RhoA Activity at Cell-Cell Junctions through Desmoplakin: Implications for Carvajal Disease."

Reviewer comments: *The Ect2 perinuclear staining is interesting (or odd?). While the authors suggest that the IF signal may be nonspecific, it appears to decrease in the Ect2 and DP KD cells. Is there evidence for nonspecific binding by western (i.e., another prominent band)? Given the recent evidence for desmosome-ER interactions (Bharathan et al., 2023), I wouldn't be quick to dismiss this staining pattern. It could reflect some interesting biology!*

Response: We agree with the reviewer that the perinuclear staining could be meaningful. Given that we have not directly demonstrated that this staining is non-specific, we have removed this wording and allude to the issue as an open question.

Reviewer comment: *Fig1 B': the linescans are not informative and the difference in colocalization is not readily apparent. At a minimum, Pearson's correlation coefficient analysis should be used to quantify and statistically compare colocalization.*

Response: We appreciate the reviewer's suggestion that the linescans are not providing added information about localization of DP with Ect2 and have opted to remove them from Figure 1.

Reviewer comment: *Fig1. C': Why not compare the average of all three points? Also, how were the stats done?*

Response: We have now included a bar graph in the figure comparing the average of the three points, with different symbols to indicate the pairs in each "N". Statistics were done using a paired t-test and we have included the statistics information in the figure legend.

Reviewer comment: *In Fig 2A, Fig 4A, and Fig6 C, I dislike how the Pg and DAPI signals were combined in blue and overlaid with the grayscale Ect2 signal. I found it confusing. Why merge the DAPI with Pg? Why blue rather than making Ect2 green and Pg red (or better, magenta) and showing the merge (minus DAPI)? Also, why is DP green rather than greyscale? The green color is unnecessary.*

Response: We appreciate the suggestions the reviewer made to present the figures in a manner that makes it easier to see localization of all the proteins. For the proximity ligation assay (PLA) results in Figure 1C we changed the color for Pg from blue to red to better highlight the cell-cell borders while the PLA spots are in white and DAPI in blue. The other figures containing DP in green have been changed to greyscale (Figures 2A, 4A, 6C) and the overlays of Pg with Ect2 or RhoA have been removed; however, greyscale images are included for all channels (Figures 2A, 4A, 6C).

Reviewer comment: *Fig 4B': it would be helpful to show examples of both "high" and "low" DP at the ICD.*

Response: We have added an example of medium DP for the DP KO (in orange) to go with "high" (red) and "low" (yellow). These examples show the range of levels seen within the analysis in Supplementary 2C. Furthermore, we include quantification for the localization of Pg at IDs in the Dsp cKO showing no significant change in Supplementary 2D.

Reviewer comment: *Fig 6B: a cartoon schematic of DP11 would be helpful.*

We added a DP11 schematic along with the DPI schematic in Figure 6B.

Reviewer #2 (Comments to the Authors (Required)):

Reviewer comments: *This paper describes the role of the desmosomal protein desmoplakin in recruiting the RhoA activator (GEF) Ect2 to intercalated disks (IDs) of cardiac myocytes and junctions of cultured cardiac-derived cells. In addition, a mechanism of activation of Ect2 by aPKC is proposed, and a significant link to human pathology illustrated by examining epidermal cells isolated from patient with a cardio-cutaneous disease (Carvajal syndrome). This paper advances significantly the field, since little attention had been devoted so far to the localisation of Rho family regulators at IDs, and mechanistic insights into disease pathogenesis are rare. The data are strongly supportive for all the data of the papers which contain appropriate controls, multiple approaches, and an impeccable description.*

This Reviewer suggests a few additional experiments that could nicely complement the data:

Reviewer comments: *1. IP experiments were only carried out using human keratinocytes. Was a colP experiment attempted using cardiac tissue or cells ? This would strengthen the biochemical argument for interaction. A biochemical experiment of direct interaction is also missing (see below).*

Response: We have now included a coimmunoprecipitation experiment in HL-1 cardiomyocytes showing the presence of DP and Ect2 in co-immunoprecipitates (Supplementary Figure 1D). The signal, while low, is specific. Altogether our data indicate that DP and Ect2 are in close proximity. We have exerted caution in the revised text to ensure that a direct interaction is not implied.

Reviewer comments: *2. Since the authors observe a decrease of DP junctional labelling in cells depleted of Ect2, does inhibition of RhoA activity perturb DP localisation at junctions of cardiac-derived cells in vitro?*

Response: While our focus in this manuscript is the role of DP in recruiting RhoA through Ect2, evidence from our lab and others supports the importance of properly tuning RhoA/ROCK activity for the recruitment and maintenance of desmoplakin/desmosomes. That is, too much or too little RhoA can interfere with their organization. For instance, we showed that RhoA activation accelerated DP redistribution to desmosomes during the first hour of junction assembly, whereas sustained RhoA activity compromised desmosome plaque maturation (e.g. Godsel et al. MBoC 2010). In addition, in a paper highlighted by referee #3 that we now cite and discuss, ROCK inhibition impairs DP recruitment in cardiomyocytes that occurs in response to EGFR/Src inhibition (PMID: 36795511). Based on these observations we would predict that a threshold level of RhoA through recruitment and regulation of Ect2 activity would ensure proper tension to recruit and/or maintain desmosomes. In other words, upon inhibition of RhoA, we predict there would be a concentration dependent disruption of DP localization in cardiac-derived cells. While this would require experimental validation, given this question was not our focus we have elected not to include this line of investigation in the manuscript.

Reviewer comments: *3. The authors convincingly show that RhoA activity is altered in cells depleted of Ect2, but it would be interesting to provide additional functional assays, for example measuring strength of adhesion, cell contractility etc.*

Response: To address the functional importance of Ect2, we included new disperse assay data in Supplementary Figure 3C, C'. These data demonstrate increased fragmentation of keratinocyte cell monolayers, consistent with the idea that the junctions in Ect2 depleted cells are less robust than in controls. We also carried out experiments (N=17) to test the importance of Ect2 in collagen gel contractility assays and were able to demonstrate decreased contraction upon DP and Ect2 knockdown in cardiomyocytes using this approach. However, due to the high degree of variability we elected not to include in the paper as we felt it would take significant optimization to achieve reproducibility/significance. For the reviewers' information we have included one of these experiments here along with a graph including data where contraction was observed in the control condition. As shown in this example, contraction was reduced by DP KD or Ect2 KD and DP11 put-back showed a trend toward rescue. While the data were significant when comparing Ctl KD and DP KD or Ctl and Ect2 KD (Two-Way ANOVA Mixed Effects Model with the Geisser-Greenhouse correction, Dunnett's multiple comparisons test, with individual variances computed for each comparison, Ctl KD vs. DP KD: *p=0.0026, Ctl KD vs. ECT2 KD: *p=0.0320, Ctl KD vs. DP KD + DP11 rescue: p=0.1597), controlling expression levels for rescue constructs was difficult to optimize.

[Figure removed by editorial staff per authors' request]

Reviewer comment. 4. The authors say that Ect2 labelling is "completely lost" in cells from Carvajal syndrome patients (Fig. 6A). However some labelling is still detectable. What structures is Ect2 associated with in these cells?

Response: As the reviewers point out there are non-junctional structures that stain with Ect2 and N-Cad in the Carvajal patient tissue. While we are uncertain about the nature of these structures, we have revised the text to more accurately state that Ect2 junctional labeling was lost in the Carvajal tissue.

Reviewer comment. 5. The authors use a DP mutant to infer about the region of DP implicated in binding to Ect2. If direct binding occurs, the binding and the mapping of the binding site can be easily demonstrated with pulldown assays using purified bait and preys. 6. A pulldown or other experiments could also be useful to compare affinities of binding of Ect2 to either DP or the cadherin-catenin complex.

Response : We agree with the reviewers that a direct interaction between DP and Ect2 has not been demonstrated and have included new language in the text to clarify this point. Given the challenges and time investment associated with generating pure protein samples of DP and DP mutants, we elected to leave the issue of whether binding is direct to future line of investigation. Regarding the current manuscript, as mentioned above we have now included a coimmunoprecipitation experiment in HL-1 cardiomyocytes showing an interaction between DP and Ect2 in Supplementary Figure 1D.

Reviewer comment.7. *The Dsp cKO myocytes (Fig. 3B) show junctions that are much more elongated compared to WT. Has electron microscopy been done in these cells to understand better the ultrastructural/morphological effects of the conditional KO?*

Response: The reviewer makes an interesting point. We agree that in Figure 3B the junctions appear more elongated. To determine whether this is a reproducible finding across our samples, we carried out a series of junction length measurements. The data indicate that there is not a significant difference in junction length overall, so we have opted not to include these data in the revised manuscript.

Reviewer comment. *The text is very clearly written, the statistics is good, and the data are presented in a logical manner, which flows beautifully.*

Minor points: - calcium medium, not media (medium is singular, media is plural in latin).

Response : We have corrected this in the manuscript.

Reviewer #3 (Comments to the Authors (Required)):

Reviewer comment : *Zarkoob et al. investigated the functional interaction between the RhoGEF Ect2 and the desmosomal component desmoplakin (Dsp), focusing on the intercalated disc (ID) and linking their findings to Carvajal disease. The authors demonstrated the interdependence of Ect2 and Dsp, with Rho as a regulatory component and involvement of PKC.*

Overall, this study provides important new insights into the composition and function of ID components and is of interest to both the cell biology and cardiology fields. To fully realize the potential of this study, we suggest the following:

Major Comments:

Reviewer comment 1. *Specificity of the effects to desmoplakin: The authors propose Ect2 function and localization to be dependent on Dsp, but how specific are the observed effects to Dsp, as opposed to general desmosomal disruption, given that Dsp is the major stabilizing component of desmosomes? The authors primarily used plakoglobin (Pg) for counterstaining; however, Pg is also a component of adherens junctions (AJs). To distinguish the role of Dsp from desmosomal components, it would be important to assess the dependence of Ect2 localization and function on other desmosomal proteins such as desmoglein-2 (Dsg2) or plakophilin-2 (Pkp2).*

Response: The reviewer raises an important point. In fact, in previous work we demonstrated that PKP2 is essential for DP recruitment to cell borders in epithelial cells (Godsel et al. 2010 MBoC) and cardiomyocytes (Dubash et al. 2016 JCB). When PKP2 is silenced, DP is mislocalized (keratinocytes) and/or expression reduced (cardiomyocytes), and it follows that Ect2 would also be mislocalized. In epithelial cells DP mis-localization coincides with increased global RhoA activity consistent with the idea that DP:Ect2 interactions may be taking place

aberrantly when not properly localized to cell-cell junctions. And as we have shown in this manuscript, loss of DP at borders, which also occurs when PKP2 is silenced, results in failure of Ect2 to accumulate at cell-cell junctions. Given the existing strong evidence that loss of PKP2 disrupts DP localization we have not repeated those experiments here. With respect to Dsg2, it has been reported that DP and PKP2 become mis-localized in end stage heart disease but otherwise are largely unchanged (remain at cell-cell junctions). This still leaves open the question of whether Dsg2 collaborates with DP to ensure Ect2 localization, and while we have not experimentally tested this, we have included discussion of this point in the revised text.

Reviewer comment 2. *Functional Link to Carvajal Mutation: The authors associate the localization of Ect2 at the ID with the Carvajal-associated Dsp mutation. Is Rho activity as a consequence then reduced by the Dsp mutation? Additionally, is Dsp still detectable at the ID in patient samples?*

Response: Our work with the keratinocytes isolated from a Carvajal patient suggests that DP remains at cell borders in vitro but with reduced ability to recruit Ect2 to the membrane, so we expect that RhoA distribution and activity would be reduced but not completely lost (Supplementary figure 3). Given our limited ability to assess this experimentally with the proper Ect2 controls (see explanation above in response to reviewer #1 regarding limitations of Ect2 antibodies), we have not directly tested the extent to which Rho is affected in these cells. While we currently don't have access to additional patient material, we expect that as previously reported in advanced disease DP is lost at borders (along with Ect2).

Reviewer comment 3. *Ect2 Reduction upon Dsp Knockdown: The authors report reduced localization of Ect2 at the ID following Dsp knockdown. Is this due to impaired recruitment of Ect2 to the ID, or does Dsp influence Ect2 expression (at the RNA/protein level) or its stability/turnover?*

Response: In the current study Ect2 protein levels do not decrease in the presence of DP knockdown (Figures 4C and 5A, B).

Reviewer comment: *Furthermore, KD of Ect2 also reduces the relative intensity of Ect2 at the ID. Would one not expect a uniform reduction in such a case?*

Response: We interpret the reviewer's comment to be directed at DP levels when Ect2 is knocked down. We agree that the levels of DP do slightly decrease when Ect 2 is silenced in the NRVCs and we have included quantification corresponding to Figure 4A in Supplementary Fig. 2B'. It is possible that Ect2 KD affects actin contraction which could impact DP trafficking and localization at cell borders (Godsel et al 2005).

Reviewer comment 4. *Immunoprecipitation Analysis (Supp 1D): The immunoprecipitation experiment is a key approach to confirm the structural association between Dsp and Ect2, supporting the data from the PLA assay. However, it was performed only in keratinocytes. To substantiate the relevance of this interaction at the ID-the central focus of the study-please provide a corresponding IP analysis in cardiomyocytes. E.g. in Fig 5B: Was Dp pulled-down with Ect2?*

Response: We have now included a coimmunoprecipitation experiment in HL-1 cardiomyocytes showing the presence of Ect2 in DP immunoprecipitates (Supplemental Figure 1D). The signal, while low, is specific, and altogether our data in epithelial and cardiac cells support the close proximity of Ect2 and DP.

Reviewer comments: *Minor Comments: Fig 1C: PLA signal appears not only at cell-cell junctions but also in the cytoplasm/nucleus-can the authors clarify this observation? Is this*

expected? Are appropriate controls for the PLA assay included (e.g., single antibody or isotype controls)?

Response: The most direct control for PLA is protein knockdown, such as the DP KD that is included as a control in Figure 1C (Hegazy et al. 2020, *Curr. Protocols*; Sharanek et al. 2022, *STAR Protocols*). While most of the signal appears to be in proximity to the membrane, we agree with the reviewer that the perinuclear staining is potentially interesting as suggested by Reviewer 1 and have included discussion of this point in the revised text.

Reviewer comments: *Fig 2A: The authors state that DP is reduced upon Ect2 knockdown. Please provide quantification of DP staining to support this claim.*

Response: We have included quantification in Supplemental Fig 2A' corresponding with data in Figure 4A to demonstrate that the levels of DP do decrease somewhat in the NRVCs (albeit only trending towards significance).

Reviewer comments: *Fig 3C: Please provide images of cardiomyocytes in a longitudinal orientation to better visualize the intercalated discs.*

Response: We assume the reviewer is referring to the Dsp cKO data. To better make the point that Ect2 is lost at junctions retaining N-cadherin staining we have placed boxes around several clearly defined IDs to demonstrate the absence of Ect2 (lower panel) in these locations. Given the issue stated above with Ect2 antibody reagents we did not stain more optimally oriented specimens, but contend that these data support the decrease in Ect2 at N-cadherin positive IDs.

Reviewer comments: *Fig 5A: A loading control for total protein after pulldown is needed to ensure equal input across conditions.*

Response: In the case of the 17A pull down experiments to capture active Ect2 everything eluted from the column is run on the gel. The lysates probed with Ect2 and GAPDH shown in the whole cell lysate blot (WCL) serve as the control to demonstrate equal loading of the columns for the assay.

Reviewer comments: *Fig 5A': The p-value indication is unclear: is the comparison made between BIM and PMA or BIM and control? What exactly does the y-axis represent? Please define the unit or measurement.*

Response: The Y axis represents the fold change of Ect2 pulled down in the 17A pulldown between control and BIM and control and PMA. P values have been added to the figure. This is clarified in the caption.

Reviewer comments: *Supp Fig 2: The y-axis label is missing-please specify what is being measured. Where were the regions of interest (ROIs) placed? Clarify how localization was determined.*

Response: We apologize for the missing axis and thank the reviewer for pointing this out. The Y axis represents all of the intensity measurements for each ROI included in the data shown in Figure 4B, B'. Border segmentation was performed on the plakoglobin channel using MaxEntropy method auto-thresholding. With segmented areas as ROIs, mean intensity was measured in each channel. This information has been added to the Materials and Methods in the section on Statistics.

Reviewer comments: *Fig 1B': Axis labels are missing from the graph. Indicate the position of the line scan in the corresponding 2D image for spatial reference.*

Response: In response to comments made by Reviewer #1 we have chosen to remove the line scans.

Reviewer comments: Fig: Which western blot does this quantification correspond to? Is it for vehicle-treated or PMA-treated samples? Clarify what is meant by "PKC:Ect2" - does this refer to a PKC phosphorylation motif?

Response: 5B' corresponds to the Western blot in 5B. It reports the ratio of the PMA-treated PKC motif signal (normalized to the vehicle treated sample) to the Ect2 that was pulled down in the immunoprecipitation. This is clarified in the caption for Fig. 5.

Reviewer comments: *Statistical Analysis: For the statistical tests applied, assessment of normal distribution and homogeneity of variance is required. Please indicate whether these assumptions were tested and met. For each experiment, provide the number of independent biological and technical replicates performed.*

Response: For each experiment we have now provided the number of independent biological replicates as well as individual ROIs and/or technical replicates tested for each experimental arm (in captions). We have also included the averages for each "N" in the plots, overlaid onto individual data points. To minimize the influence of variability and potential outliers among ROIs within a given biological replicate and to more accurately reflect the experimental design, ROI level values were averaged per each "N", and these averaged values were then used to perform paired or unpaired t-tests, as appropriate. Under the central limit theorem, the distribution of the ROI means is expected to approximate normality given the relatively large number of ROIs per biological replicate. Consequently, comparisons based on these means are expected to reasonably satisfy the normality assumption.

To assess the assumption of equal variance between groups, we additionally performed F-tests or Brown-Forsythe tests comparing variances calculated from the means of the three biological replicates. These analyses showed no significant differences in variance between groups. Therefore, the use of two sample t-tests assuming equal variances is appropriate and valid for our analyses.

Reviewer #4 (Comments to the Authors (Required)):

Reviewer comments: *The manuscript presents a novel and intriguing finding that the RhoGEF Ect2 localizes to cardiac intercalated discs (IDs) and keratinocyte desmosomes, contributing to the maintenance of an active RhoA pool at these junctions. The work addresses a significant gap in the current understanding of spatial regulation of RhoA signaling in cardiac tissue. The identification of Ect2 as the first RhoA GEF reported to localize to ID structures specifically is potentially impactful for both cardiac biology and broader cell-cell junction research. But this reviewer has few concerns regarding the experimental parts of the manuscript.*

Reviewer comments: 1. In supplementary fig 1A, the images are not aligned.

Response: We thank the reviewer for noting this error and have updated the figure to align the images.

Reviewer comments: 2. In fig 1 B', what does the axes represent? There is no clarity on how many experimental repeats were done. To demonstrate that Ect2 displays strong co-localization with DP, please quantify the co-localization of Dp with Ect2 and N-cad with Ect2.

Response: After consideration of comments posed by Reviewer #1 we have chosen to remove these line scans from the figure.

Reviewer comments: 3. In Fig. 2A, the authors describe Pg as unaltered after Dp KD, consistent with their previous study. However, the provided images do not show equal intensity of Pg between Ctrl KD and DP KD. The authors should either provide better images between CTI and DP KD or quantify the Pg and show that it was unaltered between Ctrl and DP KD. Because if Pg is altered one could not claim that DP is responsible for ect2 localization. 4. As explained above for Figure 2a, please provide better images for Pg in Fig 4a, to demonstrate that Pg is not altered in DP KD and to quantify its intensity at the IDs, thereby showing that it remains unaltered.

Response : To improve the ability to visualize Pg at membranes, we separated the channels and are now showing all in black and white in the updated figures. We assessed the intensity of Pg at the membranes for in vivo and in vitro experiments (see data in Supplemental 2 B,D corresponding with Figure 4 A and B and Figure 6C) and in all cases there is no significant difference in Pg localization between knockdown and the control. We did not include Pg intensity measurements for Figure 2A because the exposure times for Pg in these images were not collected with the intent to quantify them. Note that the Ect2 and DP exposure times were equal for all images in each “N”.

Reviewer comments: 5. Please provide a detailed explanation of the quantification methods used (e.g., cell borders/IDs, including the number of cell borders/images/IDs quantified) for each quantification performed in this manuscript). The provided information is insufficient for repetitions.

Response: A more detailed explanation of methods has been provided for all experiments and quantification methods have been included in Materials and Methods and/or the figure legends. The number of biological repeats and ROIs for each experiment has also been included in the captions.

Reviewer comments: 6. In Fig. 6c, please also quantify Pg in all conditions or provide more explicit images that show no alterations in Pg. For the reviewer, it appears that, at least in the provided images, the DPKD+DP11-FL Ect2 signal colocalized more with Pg than with DP.

Response: We assessed the intensity of Pg at the membranes in Figure 6C and in all cases there is no significant difference in Pg localization from the control. This information is provided in Figure 6C” and Pg intensity was also measured for Figure 4A and B, shown in Supplementary Figure 2B, D. We have also increased the brightness of images in 6C to better show rescued DP at borders. Note that in this experiment we were visualizing GFP remaining after fixation, rather than using an Ab to recognize GFP, possibly reducing retention of staining in some border regions. Nevertheless, we do not rule out some contribution of other desmosome components to Ect2 localization and have incorporated this point into the discussion.

Reviewer comments: 7. The authors should also discuss how the observed interaction of DP-Ect2 is essential for desmosome function in cardiomyocytes, since available literature suggests DP is necessary for ROCK-mediated desmosome assembly in cardiomyocytes (PMID: 36795511)

Response: We thank the reviewer for recommending the manuscript and have included its citation and discussion of this point in the manuscript.

February 17, 2026

RE: Life Science Alliance Manuscript #LSA-2025-03454R

Dr. Kathleen J Green
Northwestern University Feinberg School of Medicine
Laboratory of Adhesion Research Dept. of Pathology & Dermatology Northwestern University Medical School
300 E. Superior, Tarry 7-729
Chicago, IL 60611

Dear Dr. Green,

Thank you for submitting your revised manuscript entitled "RhoGEF Ect2 Supports RhoA activity at Cell-Cell Junctions through Desmoplakin". We returned this work to Reviewers 2 and 3 for their evaluation of the revisions, and both are satisfied with no further comments. We would be happy to publish your paper in Life Science Alliance pending final revisions necessary to meet our formatting guidelines.

MANUSCRIPT ORGANIZATION AND FORMATTING:

To avoid unnecessary delays in the acceptance and publication of your paper, please read the following information carefully. Full guidelines are available on our Instructions for Authors page, <https://www.life-science-alliance.org/authors>

- Please add ORCID ID for corresponding (and secondary corresponding) author - you should have received instructions on how to do so.
- Please add a Summary Blurb/Alternate Abstract in our system.
- Please add the X and Bluesky handles of your host institute/organization, as well as your own, and/or one of the authors, in our system.
- Tables should be numbered consecutively with Arabic numerals (1, 2, 3, 4); They can be included at the bottom of the main manuscript file or be sent as separate files.
- Please consider including a "Data Availability" section placed after the Materials & Methods section. Please consult our guidelines at <https://www.life-science-alliance.org/manuscript-prep#format>
- Please add a Conflict of Interest statement to your main manuscript text.
- Please add a statement of on the approval for use of human tissues, related to primary keratinocytes and tissue staining.

LSA encourages authors to provide a 30-60 second video where the study is briefly explained. We will use these videos on social media to promote the published paper and the presenting author (for examples, see <https://docs.google.com/document/d/1-UWCfbE4pGcDdcgzcmiuJI2XMBJnxKYeqRvLLrLSo8s/edit?usp=sharing>). Corresponding or first-authors are welcome to submit the video. Please submit only one video per manuscript. The video can be emailed to contact@life-science-alliance.org

FINAL FILES:

The following items are required for acceptance.

The license to publish form must be signed before your manuscript can be sent to production. A link to the license to publish form will be available to the corresponding author only. Please take a moment to check your funder requirements.

Thank you for your attention to these final processing requirements. Please revise and format the manuscript and upload materials as soon as you are able.

Thank you for this interesting contribution to the literature. We look forward to publishing your paper in Life Science Alliance.

Sincerely,

Reviewer #2 (Comments to the Authors (Required)):

The revision and responses to my comments addressed my concerns adequately.

Reviewer #3 (Comments to the Authors (Required)):

All raised points have been sufficiently addressed.

March 2, 2026

RE: Life Science Alliance Manuscript #LSA-2025-03454RR

Dr. Kathleen J Green
Northwestern University
Pathology
303 E. Chicago Ave
303 E Chicago Ave
Chicago, Illinois 60611

Dear Dr. Green,

Thank you for submitting your Research Article entitled "RhoGEF Ect2 Supports RhoA activity at Cell-Cell Junctions through Desmoplakin". It is a pleasure to let you know that your manuscript is now accepted for publication in Life Science Alliance. Congratulations on this interesting work.

Your article will publish open access upon publication under a CC-BY license.

DISTRIBUTION OF MATERIALS:

Again, congratulations on a very nice paper. I hope you found the review process to be constructive and are pleased with how the manuscript was handled editorially. We look forward to future exciting submissions from your lab.

Sincerely,
